# Multi-omics analysis and single-cell sequencing revealed the lysosome associated molecular subtypes and prognostic model development of papillary thyroid carcinoma

Jianhua Zhang[1,2,3,4], Kai Yue[1,2,3], Yansheng Wu[1,2,3], Chao Jing[1,2,3], Xudong Wang[1,2,3]*

1 Department of Head and Neck Oncology, Tianjin Medical University Cancer Institute and Hospital, Tianjin, China, 2 Tianjin Key Laboratory of Basic and Translational Medicine on Head & Neck Cancer, Tianjin Key Laboratory of Cancer Prevention and Therapy, Tianjin, China, 3 Tianjin Cancer Institute National Clinical Research Center of Cancer, Tianjin, China, 4 Department of Thyroid Surgery, The Affiliated Hospital of Qingdao University, Qingdao, China

* thyroidzhang@163.com

## Abstract

Papillary thyroid carcinoma (PTC) is the most common endocrine carcinoma in recent years, necessitating more precise risk stratification to accurately identify low-risk patients. Although preliminary evidence exists, studies on lysosomes in PTC are limited. This study utilized multi-omics data from the TCGA database to comprehensively investigate the genomic and biological characteristics of lysosomes in PTC patients and identify lysosome-associated genes (LAGs) linked to PTC prognosis. We developed a LAG scoring system for risk stratification based on the expression levels of risk coefficients and independent prognostic LAG variables. Clinical value was assessed through immune infiltration analysis, pathological subgroup analysis, immunotherapy response, and drug sensitivity prediction. Single-cell sequencing from the GEO database was used to analyze PTC samples, and bioinformatics findings were validated using western blot, qRT-PCR, colony formation, and Transwell assays. A new LAG scoring system was developed based on five prognostic LAGs, with single-cell sequencing revealing their expression in different cell types. The role of one LAG, *DNASE2B,* in PTC cell cloning, proliferation, and invasion was further confirmed *in vitro*. This comprehensive study highlights the complex interactions between lysosomes and PTC biology, offering new insights into the role of lysosomes in PTC and identifying potential targets for intervention.

## Introduction

Thyroid cancer has become the most rapidly increasing endocrine malignancy in recent years, with papillary thyroid carcinoma (PTC) being the most common pathological type, accounting for 90% of all thyroid cancers [1,2]. PTC generally has a

**Data availability statement:** The datasets used and analyzed in this study were downloaded from the publicly available TCGA database (https://portal.gdc.cancer.gov/).

**Funding:** The author(s) received no specific funding for this work.

**Competing interests:** The authors have declared that no competing interests exist.

favorable prognosis [3]. As a result, there is growing support for adopting relatively conservative treatment measures for patients with low-risk PTC [4]. However, there is still no consensus on how to more accurately identify low-risk PTC [5,6]. Additionally, aggressive variants of PTC and advanced TNM staging continue to pose threats to the quality of life of affected patients [7,8]. Therefore, more precise risk stratification for PTC is needed.

As a membrane-bound organelle, lysosomes contain highly acidic lumens that capable of breaking down complex macromolecules such as glycoproteins, glycolipids, and nucleotides [9]. Moreover, recent evidence suggests that lysosome is involved in tumor-related biological functions, including energy homeostasis regulation, cell growth, mitotic signaling, angiogenesis, and activation of transcriptional programs [10,11]. Given the increasing recognition of the role of lysosomes in tumor progression, therapies targeting lysosomes have demonstrated preliminary clinical efficacy [12]. In PTC, studies have shown that transcription factor E3 (TFE3) regulates the autophagy-lysosome pathway, driving tumor invasion and metastasis [13]. Furthermore, neutralizing the acidic environment of the lysosomal lumen can induce necrosis in thyroid cancer cells [14]. Despite this preliminary evidence, research on lysosomes in PTC remains limited, and the functional mechanisms and therapeutic targets involving lysosomes warrant further investigation.

In this study, using multi-omics data from The Cancer Genome Atlas (TCGA), we comprehensively examined the genomic and biological characteristics of lysosome-related genes in PTC patients and identified lysosome-associated genes (LAGs) relevant to PTC prognosis, subsequently established a LAG scoring system. Single-cell sequencing analysis was also employed to assess the expression patterns of the selected LAGs across different cell types. Among the identified LAGs, *DNASE2B*, which had the highest correlation coefficient, was confirmed to influence the proliferation and invasive capacity of PTC cells through *in vitro* experiments, thereby verified the reliability of bioinformatics analysis. Our study of the complex interactions between lysosome and PTC biology gave new insights into the role of lysosomes in PTC, while also provided potential therapeutic targets for intervention.

## Materials and methods

### Data collection and pre-processing

Using the TCGA database, we collected and downloaded the RNA-seq data (Platform: illumina HiSeq) along with clinical baseline characteristics (clinicopathologic features and overall survival rate), including both normal and PTC samples. In the Perl environment, we extracted and preprocessed the transcriptome matrix of the samples, annotating gene labels based on human genome features. Excluding samples lacking clinical survival outcomes and prognostic information, a total of 59 normal samples and 500 PTC samples were included for subsequent analysis (Supplementary Table 1 in S1 Table). Furthermore, due to the significant disparity in the number of M-stage samples, they were not included in the analysis of this study. Based on the MSigDB database (Molecular Signatures Database), we searched for

lysosome-related genes using "lysosome" as the keyword, identifying 161 lysosomal signatures for further analysis [15] (Supplementary Table 2 in S2 Table). In the R environment, we employed the "limma" R package to perform statistical analysis of differential expression of lysosome-associated gene (DE-LAG) signatures between normal and PTC samples with the filtering criterion of *p*.adjust (FDR) < 0.05 was applied. The *p*-values were then adjusted for multiple testing using the Benjamini-Hochberg (BH) method to control the false discovery rate (FDR).

## Construction of LAG score related prognostic model in PTC

Based on the expression levels of DE-LAG signatures and overall survival (OS) rate of PTC samples, we performed univariate Cox analysis using the "survival" R script to evaluate the prognostic value of each gene signature in PTC. Using the "glmnet" R script, we applied the LASSO (Least Absolute Shrinkage and Selection Operator) function to select LAG prognostic variables [16]. The complexity of the model is controlled through a penalty term to prevent overfitting in the LASSO regression model. The regularization parameter λ (lambda) for the model is selected using 10-fold cross-validation to determine the optimal value of λ. Through multivariate Cox analysis, we assessed the independent prognostic value of each LAG variable and calculated the risk coefficients for these variables. Based on the risk coefficients and the expression levels of LAG independent prognostic variables, we constructed an LAG scoring system for risk stratification in PTC samples. The LAG scoring formula was: *LAG score = NPC2 x −0.0033 + CTSV x 0.6578 + DNASE2B x 2.0785 + SLC11A1 x 0.7991 + NEU4 x 1.5434.* According to the optimal survival cut-off value (the cutoff value with the maximum log-rank test) in PTC samples, we divided the PTC samples into two LAG score subgroups and plotted the clinical survival curves for different LAG score subgroups using the "Survival" R script.

## Identification of LAG molecular subtype landscape and recognition of immune microenvironment infiltration features

Using the "ConsensusClusterPlus" R script, we applied the "K-means" algorithm to identify the LAG molecular subtype landscape in PTC samples based on the expression characteristics of LAG independent prognostic signatures [17]. According to the optimal model classification from consensus clustering analysis (k = 2~9), we divided PTC samples into two LAG molecular subtypes. The "ggplot2" R script was used to generate a principal component analysis (PCA) plot to display the distribution characteristics of the LAG molecular subtypes [18]. Using the reference gene list from KEGG signaling pathways (c2.cp.kegg.v7.2.symbols.gmt), we performed the "GSVA" analysis algorithm (min.sz = 10, max.sz = 500, parallel.sz = 1) to evaluate differentially regulated KEGG pathways between the LAG molecular subtypes [19]. Based on the transcriptome characteristics of PTC samples, we conducted an ESTIMATE analysis using the "estimate" R script to quantitatively assess the immune infiltration status of each sample, including immune scores, stromal scores, and ESTIMATE scores [20]. Using the marker genes of 23 immune cell types, we applied the ssGSEA evaluation algorithm with the "GSVA" R script to calculate the infiltration proportions of these 23 immune cells in each sample [21].

## Consistency validation of the LAG scoring system and clinical pathological subgroup analysis

To validate the stability and independence of the LAG scoring system in predicting the clinical prognosis of PTC, we used the "caret" R script to divide PTC samples into two independent cohorts: a training set and a validation set, with a 1:1 ratio. Based on the risk scores and expression profiles of the LAG prognostic signatures in both cohorts, we classified the PTC samples into two LAG score subgroups and plotted survival curves using the "survival" R script. Time-dependent ROC curves were generated using the "survivalROC" R script to calculate the AUC values for 1, 3, and 5 years. A Sankey diagram was drawn using the "ggalluvial" R script to demonstrate the relationship between LAG molecular subtypes, the LAG scoring system, and PTC clinical prognosis. Based on the clinical-pathological characteristics of the PTC samples and the LAG scoring system, we developed a nomogram diagnostic model using the "rms" R script to predict the 1-, 3-,

and 5-year survival probabilities of PTC samples. Additionally, survival curves for different clinical-pathological subgroups of PTC were plotted using the "survival" R script.

## Preprocessing and analysis of single-cell RNA sequencing data in PTC

The single-cell RNA sequencing data for PTC used in this study was obtained from the GEO database. Based on the GSE184362 dataset, we extracted single-cell sequencing data from 7 PTC samples for subsequent analysis. First, we imported and preprocessed the data using the "Seurat" R package in the R environment. To improve data quality, we normalized the gene expression matrix for each sample and filtered out low-quality cells. The filtering criteria were as follows: cells with fewer than 200 detected genes or with more than 5% mitochondrial gene content were removed. After data cleaning, we performed normalization using the "LogNormalize" method and using the "FindVariableFeatures" function to identify the top 2000 highly variable genes from the expression matrix and performed principal component analysis (PCA), selecting the top 20 principal components for further analysis. To eliminate batch effects, the Harmony algorithm was used for batch effect correction. The batch information was provided as an input variable to the "RunHarmony" function to remove batch effects. The batch-corrected data were then used for subsequent dimensionality reduction and clustering analysis. Dimensionality reduction was performed using "RunUMAP" and "RunTSNE" and clustering of cells was conducted using the "FindClusters" function with a resolution of 1.2 to define cell populations. The dimensionality reduction and visualization were performed using tSNE and UMAP (Uniform Manifold Approximation and Projection) to display the spatial distribution of different cell populations [22]. To further investigate the characteristics of different cell subpopulations, we employed the "FindMarkers" function to identify differentially expressed genes between cell populations. The selection criteria included a *p*-value threshold of less than 0.05, with FDR correction applied (Supplementary Table 3 in S3 Table). Cell types were annotated using the "SingleR" package based on the Human Primary Cell Atlas dataset, followed by manual annotation refinement using the CellMarker database.

## Prediction of immunotherapy response and drug sensitivity

Using the Cancer Immunome Database (TCIA), we obtained comprehensive immunogenomic analysis results for PTC samples and analyzed the differential expression of immunophenoscore (IPS) in different subgroups using the "limma" R script [23]. Additionally, based on the transcriptome characteristics of PTC samples and the Genomics of Drug Sensitivity in Cancer (GDSC) database, we used the "pRRophetic" R script to evaluate the response of different subgroups to small-molecule compounds by estimating their IC50 values [24].

## Cell culture

The Nthy-ori 3–1 (simian virus SV-40 immortalized normal human thyroid epithelial cell line, Sigma-Aldrich, USA) and TPC-1 (thyroid papillary carcinoma cell line, ATCC, USA) cell lines were cultured in different media. Nthy-ori 3–1 cells were maintained in RPMI-1640 medium supplemented with 10% fetal bovine serum (FBS, Gibco, USA) and 1% penicillin-streptomycin solution (Penicillin-Streptomycin, Gibco, USA). TPC-1 cells were cultured in Dulbecco's Modified Eagle Medium (DMEM, Gibco, USA) containing 10% fetal bovine serum and 1% penicillin-streptomycin solution. All cells were incubated in a 37°C, 5% $CO_2$ incubator (Thermo Fisher Scientific, USA). When the cell density reached 80%−90%, cells were digested with 0.25% trypsin-EDTA solution (Gibco, USA) and passaged at a ratio of 1:3. To ensure optimal cell conditions, the culture medium was changed every 2–3 days, and cell morphology was regularly monitored using a DMIRB inverted microscope (Leica Microsystems, Wetzlar, Germany).

## Western blot analysis

Nthy-ori 3–1 and TPC-1 cells were cultured until they reached 80%−90% confluence, then digested with 0.25% trypsin and collected. After washing the cells twice with phosphate-buffered saline (PBS), they were lysed in RIPA lysis buffer

(Beyotime, China) containing protease inhibitors for 30 minutes to extract total protein. The lysate was centrifuged at 12,000 rpm for 15 minutes (4°C), and the supernatant was collected to determine the protein concentration using a BCA protein assay kit (Thermo Fisher Scientific, USA). Equal amounts of protein samples (30 µg per well) were separated by SDS-PAGE gel (10%) and transferred to a PVDF membrane (Millipore, USA). 0.05% TBST buffer for Western blot is prepared by adding 0.5 mL of Tween 20–1 liter of Tris-buffered saline (TBS). The membrane was blocked with TBST buffer containing 5% non-fat dry milk for 1 hour at room temperature, followed by overnight incubation at 4°C with primary antibodies against DNASE2B (1:1000, PA5–31391, Thermo Fisher Scientific, USA) and the internal control GAPDH (1:5000, 14C10, Cell Signaling Technology, USA). The membrane was washed three times with TBST for 10 minutes each time, and then incubated with TBST containing 5% non-fat milk diluted HRP-conjugated secondary antibody (1:5000, Abcam, UK) at room temperature for 1 hour. After washing, protein signals were detected using enhanced chemiluminescence (ECL) reagent (Thermo Fisher Scientific, USA), and the band intensity was captured and analyzed (ChemiDoc MP, Bio-Rad, USA). The relative expression levels of protein bands were quantified using ImageJ software, with GAPDH serving as an internal control for normalization.

## Construction of DNASE2B knockdown cell model

DNASE2B gene knockdown in TPC-1 cells was achieved through transfection with small interfering RNA (siRNA). Specific siRNA sequences targeting the DNASE2B gene (si-DNASE2B) and a negative control (si-NC) were designed and synthesized by GenePharma (Shanghai, China). When TPC-1 cells reached 50%−60% confluence, transfection was performed using Lipofectamine 3000 transfection reagent (Invitrogen, USA) according to the manufacturer's instructions. Each well received 50 nM of siRNA and an appropriate amount of transfection reagent, and the cells were incubated at 37°C in a 5% $CO_2$ incubator for 48 hours. After 48 hours of transfection, the cells were collected, and Western blot analysis was conducted to verify the protein expression level of DNASE2B, assessing the knockdown efficiency. Specific operational steps were referenced in the qPCR and Western blot sections. The experimental groups included the si-DNASE2B group and the si-NC group, with all experiments repeated three times.

## Colony formation and transwell invasion assays

After DNASE2B gene knockdown in the TPC-1 cell line, colony formation assays were performed to evaluate cell proliferation ability. First, TPC-1 cells transfected with si-DNASE2B and si-NC were digested, counted, and seeded at a density of 500 cells/well in 6-well plates, with three replicates for each group. The cells were cultured in DMEM medium containing 10% fetal bovine serum (FBS) at 37°C in a 5% $CO_2$ incubator. At the end of the culture period, the medium was discarded, and the cells were gently washed with PBS, followed by fixation with 4% PBS diluted paraformaldehyde for 20 minutes. Subsequently, the cells were stained with 0.1% crystal violet solution (Beyotime, China) for 15 minutes, washed with water to remove excess dye, and allowed to dry. Clonal images were captured using an inverted microscope, and the number of colonies per well was calculated and analyzed. To assess the invasive ability of TPC-1 cells after DNASE2B gene knockdown, Transwell invasion assays were conducted. Matrigel (Corning, USA) was first diluted 1:8 and evenly coated on the membrane surface of the Transwell chamber (8.0 µm pore size, Corning, USA), incubated at 37°C for 30 minutes to allow solidification. Transfected TPC-1 cells (si-DNASE2B and si-NC) were digested, counted, and seeded at a density of $5 \times 10^4$ cells/well in the upper chamber, using serum-free DMEM medium. DMEM medium containing 10% FBS was added to the lower chamber as a chemical attractant. The cells were incubated at 37°C in a 5% $CO_2$ incubator for 24 hours. After incubation, non-invading cells in the upper chamber were discarded, and the remaining cells were washed twice with PBS, followed by fixation with 4% paraformaldehyde for 20 minutes. The cells were then stained with 0.1% crystal violet solution for 15 minutes, and cells in the upper chamber were wiped away with cotton swabs. Five random fields were selected using an inverted microscope to count and analyze the number of invading cells.

## MTT cell proliferation assay

To evaluate the proliferation ability of TPC-1 cells, an MTT assay was performed. First, TPC-1 cells transfected with si-DNASE2B and si-NC were digested, counted, and seeded at a density of $2 \times 10^3$ cells/well in 96-well plates, with five replicates for each group. The cells were incubated in a 37°C, 5% $CO_2$ incubator for 24, 48, 72, and 96 hours. At each time point, 20 μL of a 5 mg/mL MTT solution (Sigma-Aldrich, USA) was added to each well, and the incubation continued for another 4 hours. After incubation, the supernatant was discarded, and 150 μL of dimethyl sulfoxide (DMSO, Sigma-Aldrich, USA) was added to dissolve the purple crystals. The plates were then shaken at 37°C for 10 minutes. The optical density (OD) of each well was measured at a wavelength of 570 nm using a microplate reader (Multiskan FC, Thermo Fisher Scientific, USA). The proliferation of each group of cells was calculated by comparing the OD values with those of the control group, and the results were presented as the mean ± standard deviation (mean ± SD).

## Statistical analysis

We conducted data preprocessing and visualization of the raw data from PTC samples using R and Perl languages. Statistical comparisons between two groups were performed using the Wilcoxon rank-sum test, while comparisons among multiple groups were analyzed using one-way ANOVA. Log-rank test method was conducted to evaluate the clinical survival outcome for samples between different subgroups. The data from cellular experiments were evaluated through three independent replicates. All statistical differences in this study were subjected to multiple testing corrections, with a significance level set at $p < 0.05$. In this study, we applied the Benjamini-Hochberg (BH) method to adjust $p$-values in order to control the False Discovery Rate (FDR). The significance levels are indicated as follows: *$p < 0.05$; **$p < 0.01$; ***$p < 0.001$.

# Results

### Differential analysis of lysosome associated genes and construction of prognostic scoring model in PTC

In this study, we extracted 161 lysosomal gene signatures to elucidate their potential regulatory roles in PTC. By setting the criteria for differential expression as $p$.adjust (FDR) < 0.05, we defined the differential expression of LAG in PTC and identified 126 differentially expressed LAG signatures (DE-LAG) (Fig 1A). By integrating the clinical survival characteristics of PTC samples, we assessed the potential association between the 126 DE-LAG signatures and clinical prognosis in PTC. Univariate Cox analysis indicated that 15 DE-LAG signatures were significantly associated with clinical survival outcomes in PTC, comprising 11 risk factors and 5 protective factors (Fig 1B). Through LASSO analysis, we identified 13 LAG feature variables that were correlated with prognosis in PTC (Fig 1C). Based on the multivariate Cox analysis algorithm, we calculated the risk scores for each independent prognostic variable and developed a LAG scoring system for risk stratification in PTC (Fig 1D). The results from clinical survival curves indicated that within the LAG scoring system, the OS rate for PTC samples in the high LAG score subgroup was significantly lower than that in the low LAG score subgroup, suggesting that PTC samples in the low LAG score subgroup may experience better clinical survival outcomes (Fig 1E).

### Identification of LAG-related molecular subtype characteristics in PTC

Based on the expression profiles of LAG as an independent prognostic factor, we evaluated the molecular subtype characteristics of LAG in PTC using an unsupervised consensus clustering analysis algorithm. According to the optimal classification ratio of the consensus clustering algorithm (k = 2), we accurately classified the PTC samples into two distinct molecular subtypes (Fig 2A–C). Additionally, the results of the PCA plot showed a clear separation pattern of PTC samples between the LAG molecular subtypes, suggesting a high degree of independence between them (Fig 2D). Clinical survival curve analysis indicated that patients in LAG subtype B had significantly worse clinical outcomes compared to those in LAG subtype A (Fig 2E). Using GSVA analysis, we preliminarily assessed the differentially regulated KEGG

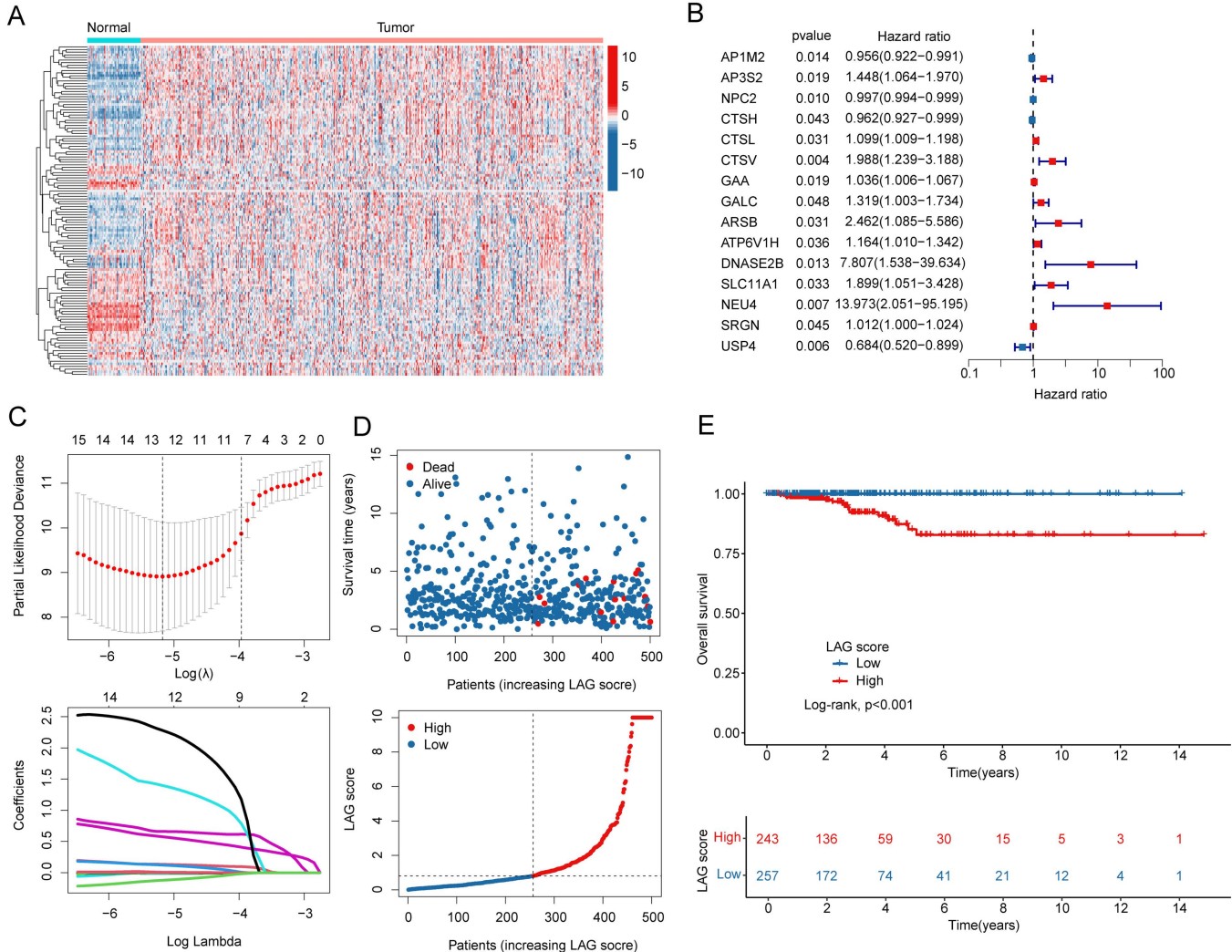

**Fig 1. Differential Expression Characteristics of Lysosome Signatures in PTC and Construction of the Prognostic Scoring System.** (A) Differential expression analysis of lysosomal genes between PTC samples and adjacent normal samples, with blue representing downregulation and red representing upregulation. The thresholds for differential expression are |Fold change| > 1 and $p$.adjust <0.05. (B) Univariate Cox analysis identifying LAG signatures associated with PTC prognosis. (C) LASSO analysis selecting prognostic LAG signature variables. (D) Construction of the LAG prognostic scoring system in PTC samples. (E) Clinical prognostic curve analysis for the LAG scoring subgroups.

signaling pathways between the LAG molecular subtypes, which may explain the key mechanisms contributing to the differences in clinical outcomes. Differential analysis revealed that in LAG subtype B, the Lysine degradation and mTOR signaling pathways were significantly upregulated, while in LAG subtype A, pathways associated with the lysosome and various metabolic processes were significantly upregulated, including alpha-linolenic acid, arachidonic acid metabolism, pyrimidine metabolism, and glutathione metabolism (Fig 2F). We noted a potential link between type 2 diabetes and LAG subtypes, which may be associated with differences in patient prognosis. Reports indicate an increased risk of thyroid cancer in diabetic patients [25]. Meanwhile, diabetes treatment with GLP-1 drugs is associated with a lower risk of thyroid cancer, which suggests an intriguing correlation [26].

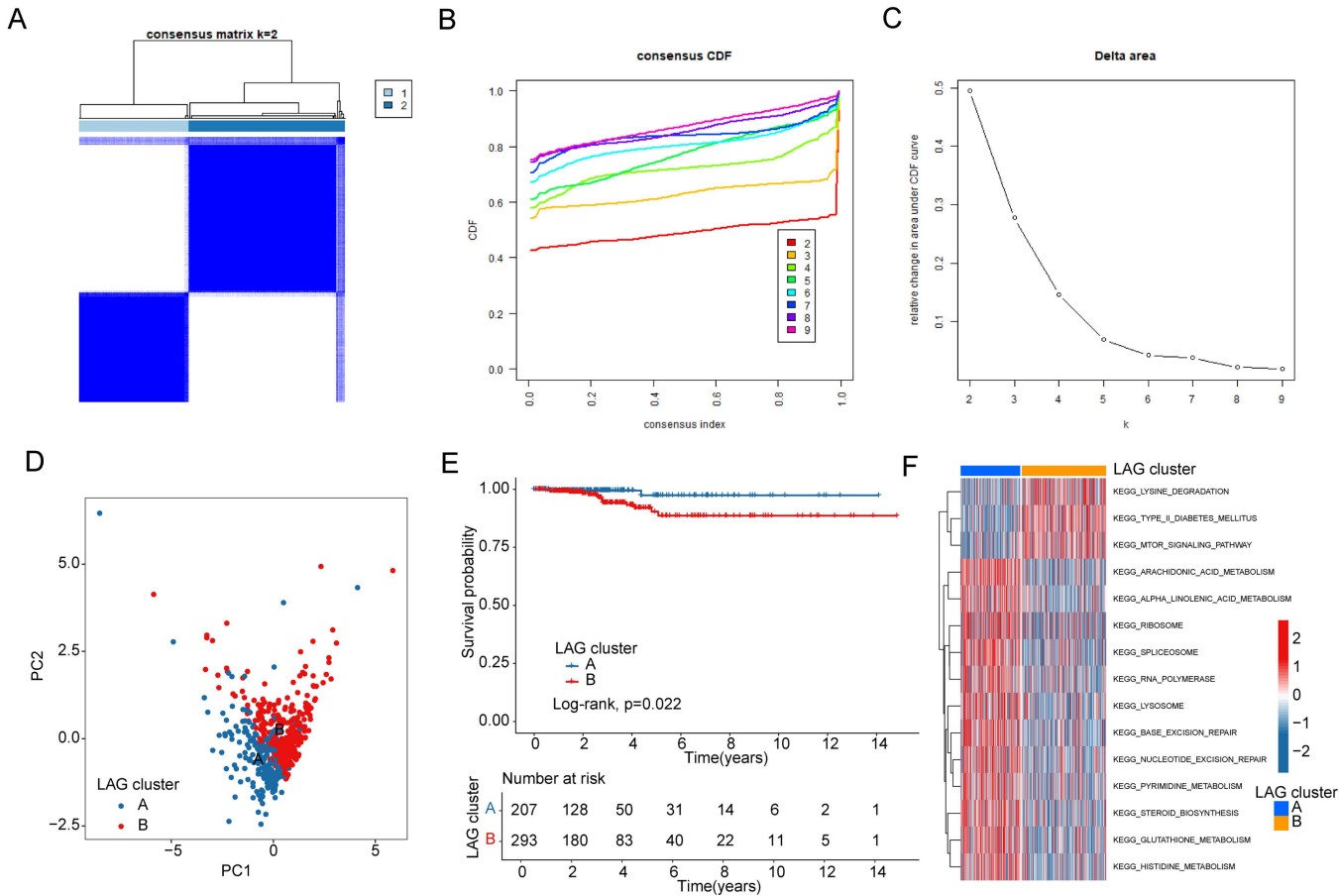

**Fig 2. Identification of LAG Molecular Subtype Characteristics and Prognostic Analysis.** (A-C) Molecular subtype analysis of LAG in PTC samples. (D) PCA plot revealing the distribution characteristics of LAG molecular subtypes. (E) Clinical prognostic curve analysis of LAG molecular subtypes. (F) GSVA analysis showing the differential regulation of KEGG signaling pathways between LAG molecular subtypes.

## Immune microenvironment infiltration characteristics and immunotherapy response of LAG molecular subtypes

Using multiple immune infiltration assessment algorithms, we explored the immune infiltration status and predicted immunotherapy responses in PTC samples of LAG molecular subtypes. The ESTIMATE algorithm results showed that in LAG subtype B, the ESTIMATE score and immune score were significantly lower compared to LAG subtype A, suggesting that PTC samples in LAG subtype B may exhibit an immunosuppressive state (Fig 3A–C). Through the ssGSEA assessment algorithm, we quantitatively analyzed the infiltration proportions of 23 immune cell types in LAG molecular subtypes. The results indicated that most immune cell infiltration proportions were significantly downregulated in LAG subtype B, including activated CD8+ T cells, activated dendritic cells, CD56bright natural killer cells, CD56dim natural killer cells, and immature dendritic cells; In contrast, eosinophil infiltration was lower in LAG subtype A (Fig 3D). Based on data from the TCIA database, we predicted the response of PTC samples to PD1 and CTLA4 immunotherapy in LAG molecular subtypes. The IPS results indicated that compared to LAG subtype B, LAG subtype A had better positive scores for PD1 or CTLA4, suggesting that PTC samples in LAG subtype A may derive greater clinical benefit from immunotherapy (Fig 3E–G).

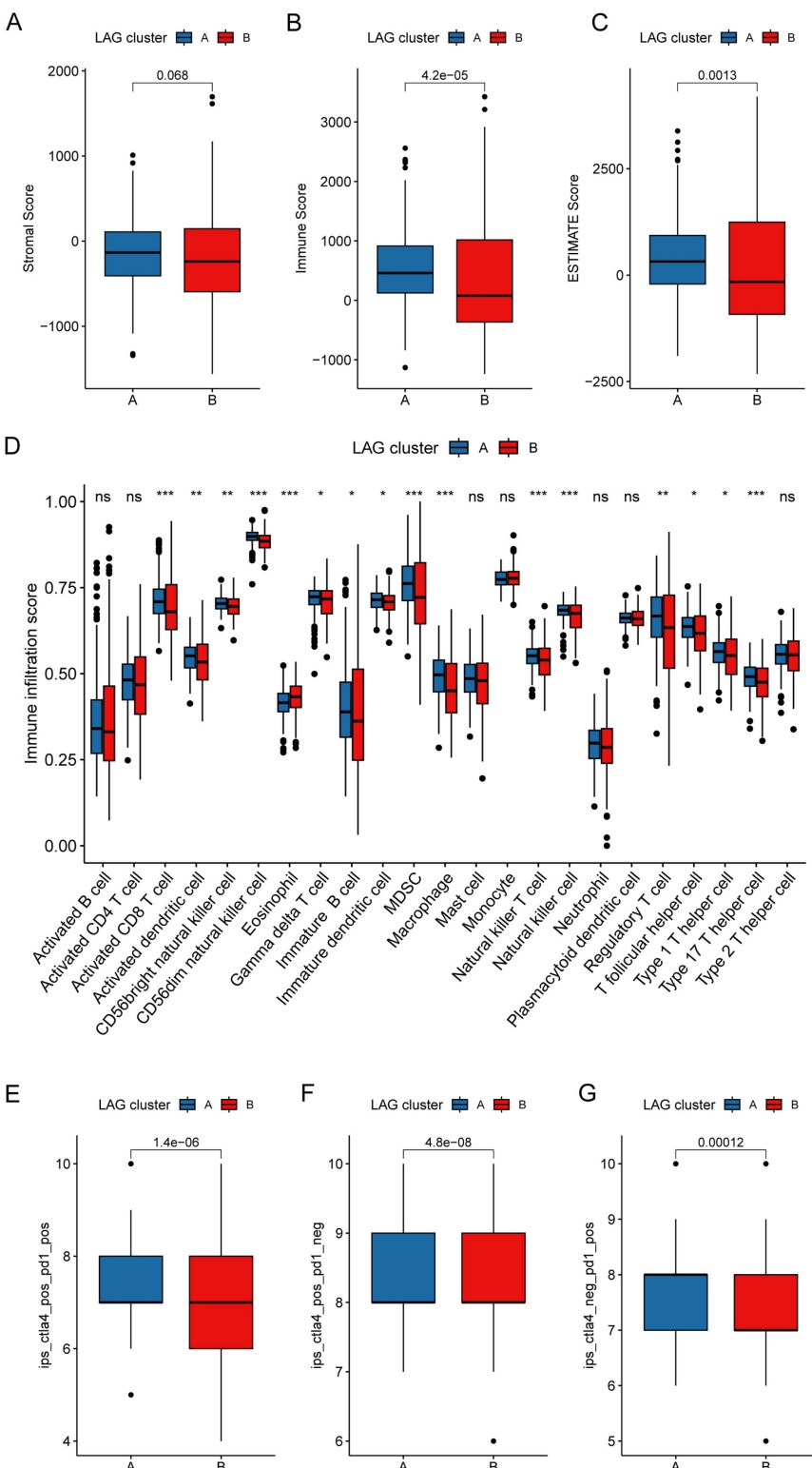

**Fig 3. Immune Infiltration Landscape and Immunotherapy Response Evaluation of LAG Molecular Subtypes.** (A-C) Quantitative assessment of immune infiltration status. (D) Proportion calculation of 23 immune cell types based on the ssGSEA algorithm. (E-G) Prediction of immunotherapy response in LAG molecular subtypes.

### Independent validation of the LAG scoring system in predicting clinical outcomes in PTC

In the subsequent research, we assessed the independence and robustness of the LAG scoring system in predicting the prognosis of PTC samples and elucidated the potential association between LAG scoring subgroups and LAG molecular subtypes. Differential analysis showed that PTC samples in the poor prognosis LAG subtype B exhibited higher levels of LAG scores (Fig 4A). Using a Sankey diagram, we detailed the close relationship between LAG molecular subtypes, the LAG scoring system, and clinical outcomes in PTC (Fig 4B). Based on the "caret" algorithm, we divided the PTC samples into two independent cohort subgroups in a 1:1 ratio to validate the independence and stability of the LAG scoring system in predicting clinical outcomes. In both the training and validation cohorts, we observed that PTC samples in the high LAG score subgroup had significantly worse clinical survival outcomes compared to those in the low LAG score subgroup, demonstrating that the LAG scoring system can accurately assess clinical survival outcomes of PTC samples (Fig 4C, D). Additionally, the time-dependent ROC curve analysis showed AUC values of 1, 0.941, and 0.910 for 1, 3, and 5 years in the training cohort, and 0.705, 0.667, and 0.821 in the validation cohort, reflecting significant diagnostic efficiency (Fig 4E, F).

### Construction of a nomogram diagnostic model based on the LAG scoring system and clinicopathological subgroup analysis

Using the clinicopathological characteristics and LAG scoring system of PTC samples, we further developed a nomogram diagnostic model to predict the 1-year, 3-year, and 5-year survival probabilities of PTC samples (Fig 5A). Time-dependent ROC curve analysis showed AUC values of 0.798, 0.840, and 0.869 for 1, 3, and 5 years in the entire cohort (Fig 5B). Moreover, the ROC curve indicated an AUC of 0.891 for the LAG scoring system, demonstrating satisfactory diagnostic performance (Fig 5C). We also assessed the distribution of LAG scores across various clinicopathological features of PTC samples, revealing significant differences in age and N stage (Fig 5D). Further, clinical subgroup survival curves demonstrated that in subgroups of age (<65, ≥65), gender (male, female), stage III-IV, and N stage (N0, N1), the high LAG score subgroup consistently exhibited poorer clinical survival outcomes (Fig 5E).

### Immune microenvironment infiltration characteristics and immunotherapy response in LAG scoring subgroups

We further evaluated the immune microenvironment infiltration landscape in PTC samples across LAG scoring subgroups. The ssGSEA assessment results indicated that in the high LAG score subgroup, the infiltration proportions of immune cells such as activated B cells, eosinophils, and neutrophils were significantly downregulated, while the infiltration of CD56dim natural killer cells was notably upregulated in the low LAG score subgroup (Fig 6A). Immunotherapy response predictions showed that the IPS scores in the low LAG score subgroup was significantly higher than those in the high LAG score subgroup, suggesting that PTC samples in the low LAG score subgroup may exhibit a better response to CTLA4 and PD1 immunotherapy (Fig 6B–D). Using the Pearson correlation algorithm, we elucidated the close relationship between the LAG prognostic signature and the immune microenvironment infiltration landscape in PTC. The results revealed that *SLC11A1* and *DNASE2B* were significantly positively correlated with 23 types of immune-infiltrating cells; *NPC2* was significantly negatively correlated with eosinophils but positively correlated with other immune-infiltrating cells; and *NEU4* was significantly negatively correlated with most immune-infiltrating cells (Fig 6E).

### Potential mechanism regulation and drug sensitivity analysis in LAG scoring subgroups

To further elucidate the potential molecular mechanisms underlying the LAG scoring subgroups, we identified differentially expressed genes (DEGs) between the two scoring subgroups with thresholds of |FC|>2 and *p*.adjust<0.05 (Fig 7A). GO enrichment analysis indicated that the DEGs between the LAG scoring subgroups were associated with biological functions such as long-chain fatty acid transport, detoxification of copper ion, stress response to copper ion, high-density

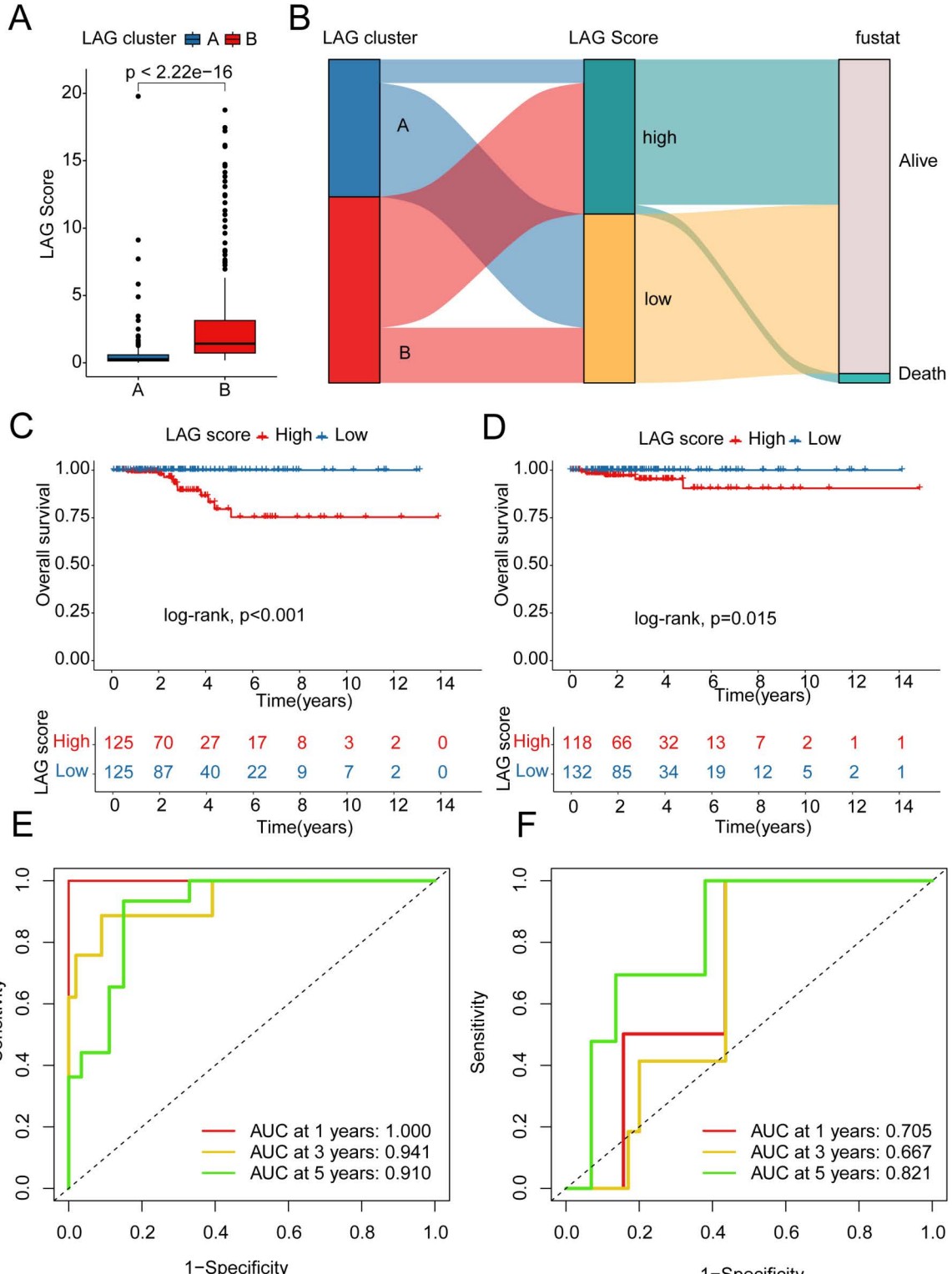

**Fig 4. Validation of the Consistency and Stability of the LAG Scoring System Model.** (A) Differential analysis of LAG scores within LAG molecular subtypes. (B) Sankey diagram analysis of the relationship between LAG molecular subtypes, LAG scoring system, and PTC survival status. (C, D) Clinical survival curve analysis of LAG scoring subgroups in independent cohorts. (E, F) Time-dependent ROC curve analysis for 1-year, 3-year, and 5-year survival in independent cohorts.

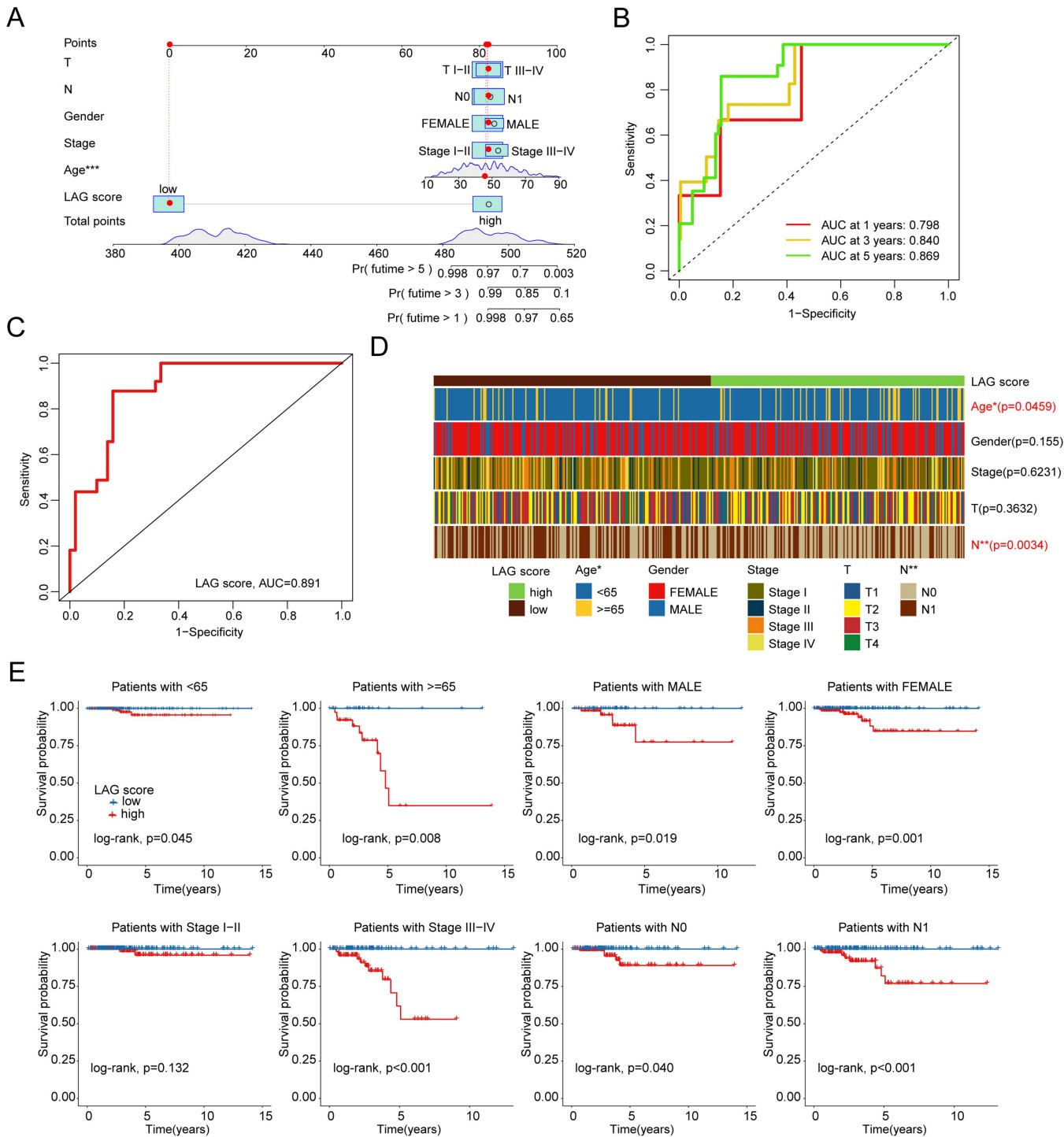

**Fig 5. Construction of a Nomogram Diagnostic Model Based on Clinical Pathological Features and LAG Scoring System, and Clinical Pathological Subgroup Analysis.** (A) Nomogram diagnostic model predicting survival probabilities of PTC at different time points based on clinical pathological variables and LAG scores. (B) Time-dependent ROC curve analysis. (C) Assessment of the diagnostic ability of the LAG scoring system. (D) Distribution of LAG scores across clinical pathological features in PTC. (E) Clinical prognostic curve analysis of LAG scoring subgroups across different clinical pathological subgroups.

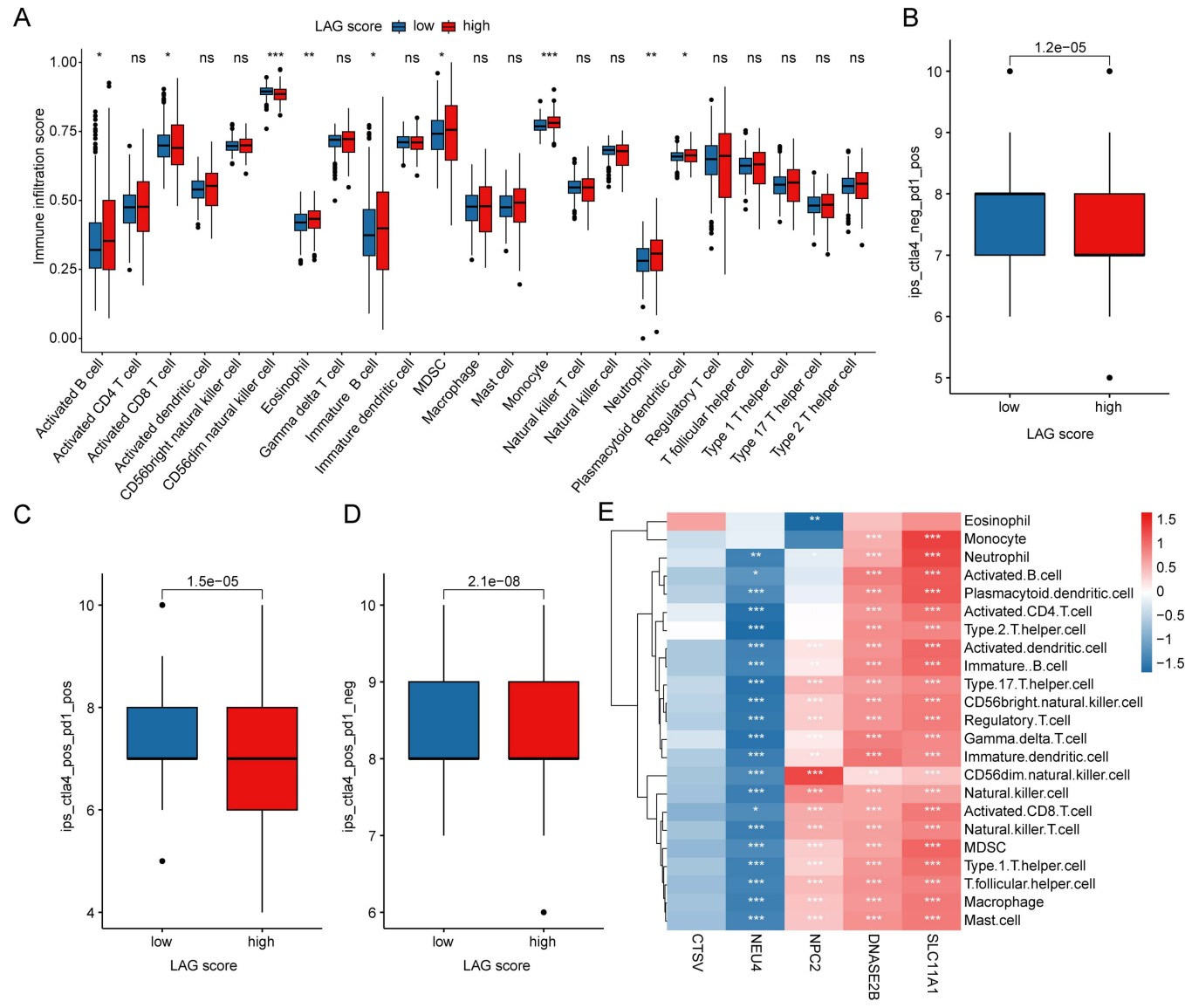

**Fig 6. Immune Microenvironment Landscape Characteristics and Immunotherapy Response Prediction in LAG Scoring Subgroups.** (A) Assessment of the proportion of 23 immune cell types based on the ssGSEA algorithm. (B-D) Prediction of immunotherapy response. (E) Correlation analysis between LAG prognostic signatures and immune infiltration characteristics.

lipoprotein particle, and plasma lipoprotein particle (Fig 7B). KEGG enrichment analysis revealed that signaling pathways like cytokine-cytokine receptor interaction, pancreatic secretion, mineral absorption, and the PPAR signaling pathway mediated the potential regulatory roles of DEGs in PTC (Fig 7C). Drug sensitivity analysis showed that the IC50 values of Doxorubicin, Etoposide, Imatinib, Parthenolide, and Roscovitine were significantly lower in the low LAG score subgroup than in the high LAG score subgroup, suggesting that PTC samples in the low LAG score subgroup may derive greater therapeutic benefits from these drugs (Fig 7D–H).

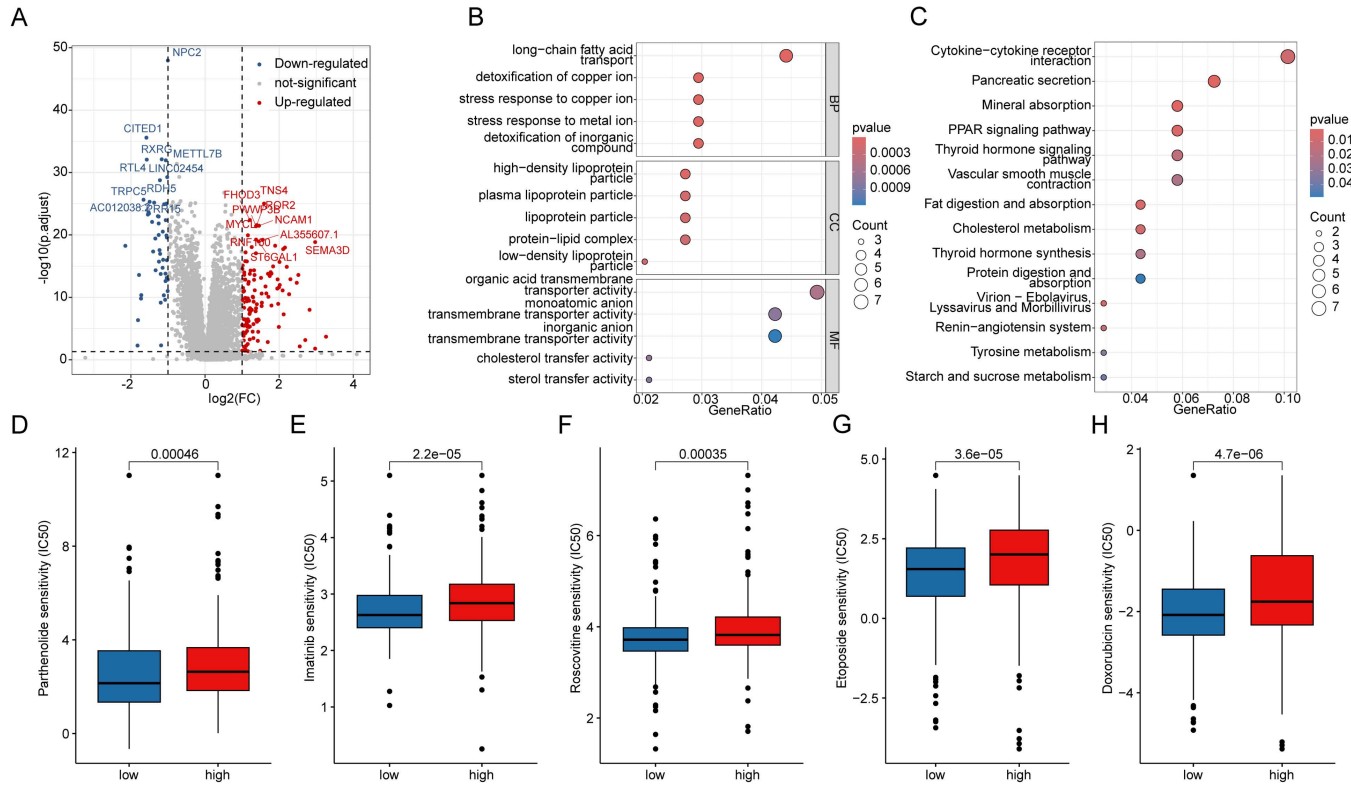

**Fig 7. Potential Mechanism Analysis and Drug Sensitivity Prediction in LAG Scoring Subgroups.** (A) Differential expression analysis of genes between LAG scoring subgroups. (B, C) KEGG and GO enrichment analysis of differentially expressed genes. (D-H) Prediction of drug sensitivity in LAG scoring subgroups.

## Single-cell sequencing analysis reveals the expression characteristics of the LAG prognostic signature in PTC cell subpopulations

In the subsequent study, we identified cell subpopulation classifications in PTC samples at the single-cell sequencing level and explored the expression distribution of the LAG prognostic signature within each cell subpopulation. Based on the GSE184362 single-cell dataset, we extracted single-cell sequencing data from seven PTC samples for further analysis. After quality control and normalization of the single-cell sequencing data for each sample, we obtained the top 2000 highly variable genes for subsequent dimensionality reduction analysis (Fig 8A, B). Using marker genes for each cell subpopulation, we employed tSNE and UMAP to display 22 distinct cell subpopulations in the PTC samples (Fig 8C, D). A visualized heatmap illustrated the expression characteristics of marker genes in each cell subpopulation (Fig 8E). Using the singleR cell annotation algorithm, we scored and annotated each cell type, identifying a total of eight cell types, including monocytes, T cells, epithelial cells, NK cells, tissue stem cells, endothelial cells, myofibroblasts, and dendritic cells (DCs). The distribution characteristics of these eight cell types in PTC samples were demonstrated using tSNE and UMAP (Fig 8F, G). Furthermore, we evaluated the expression characteristics of LAG signature genes across the eight cell types. The violin results indicated that in the seven PTC samples, the LAG signature was prominently expressed in cell types such as monocytes, T cells, and epithelial cells, while its expression was lower in DC cells (Fig 8H). Based on these findings, we have revealed the cellular subpopulation classification of PTC samples at the single-cell sequencing level and clarified the expression patterns of the LAG signature in different cell types.

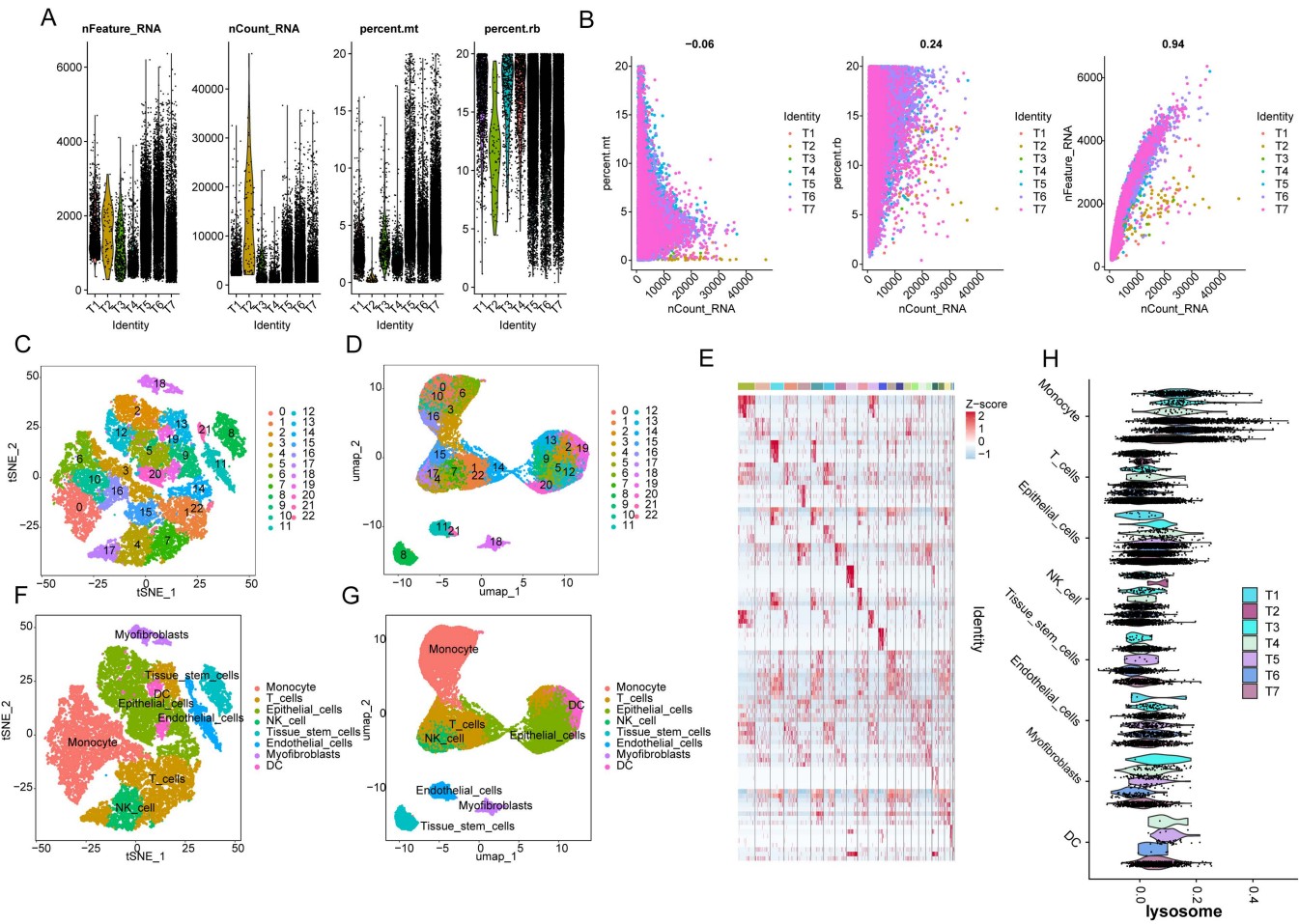

**Fig 8. Single-Cell Sequencing Analysis Reveals the Distribution Characteristics of LAG Prognostic Signatures in Cell Subpopulations.** (A, B) Quality control and normalization of single-cell sequencing data from 7 PTC samples. (C, D) t-SNE and UMAP dimensionality reduction model plots revealing the distribution characteristics of 22 cell subpopulations in PTC. (E) Heatmap analysis of marker gene expression across the 22 cell subpopulations. (F, G) Cell type annotation based on the singleR algorithm and dimensionality reduction model plot analysis (t-SNE/UMAP). (H) Violin plot analysis of LAG signatures across 8 cell types.

## DNASE2B promotes thyroid carcinoma tumor proliferation and migration

In view of the fact that *DNASE2B* had the highest HR coefficient in the screening process, we chose *DNASE2B* as the research focus to verify the reliability of the bioinformatics analysis in this study. To explore the role of *DNASE2B* played in thyroid carcinoma, Nthy-ori 3−1 were chosen as the normal thyroid epithelial cell line, and TPC-1 as human thyroid cancer model. Western blot assays demonstrated that DNASE2B were significantly upregulated in thyroid carcinoma, while siRNA effectively interfered with *DNASE2B* expression (Fig 9A–D). To examine the role of *DNASE2B* in colony formation, proliferation and invasion, we transfected TPC-1 cells with empty vector siNC or *DNASE2B* siRNA. Colony formation assays revealed that *DNASE2B* silencing decreased the mean colony number (Fig 9E,F). Transwell assay showed that knockdown of *DNASE2B* inhibited the invasion of cells compared with the cells transfected with inhibitor NC (Fig 9G,H). As shown in Fig 9I, *DNASE2B* knockout markedly inhibited cell proliferation. Collectively, our findings indicate that *DNASE2B* is highly expressed in thyroid carcinoma and plays a crucial role in the imbalance of proliferation and invasion of thyroid carcinoma cells.

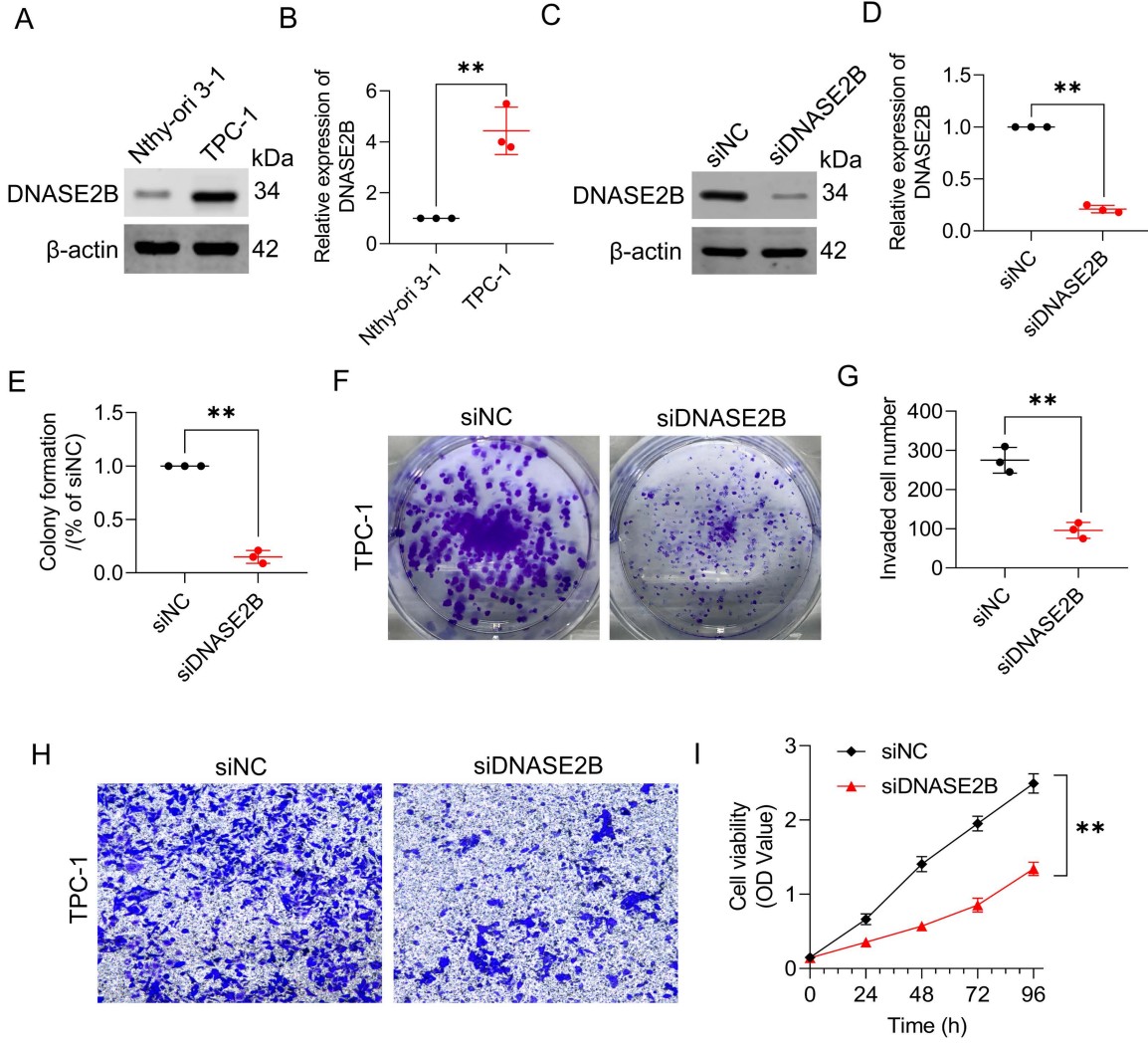

**Fig 9. DNASE2B regulated the proliferation and invasion of thyroid carcinoma cell.** (A, B) DNASE2B expression in Nthy-ori 3−1 and TPC-1 cells were analyzed by Western blot. (C, D) DNASE2B expression in TPC-1 cells was analyzed by Western blot after treatment with DNASE2B siRNA. (E, F) Colony formation assay was performed to detect the proliferation ability of cells after interfering with DNASE2B siRNA. (G, H) Cell invasion ability was decreased when transfected with DNASE2B siRNA. Cell invasion abilities were measured by a Transwell assay method. (I) MTT assay revealed that knockout of DNASE2B reduced the growth rate. *$P < 0.05$; **$P < 0.01$; ***$P < 0.001$.

## Discussion

In this study, we successfully stratified the risk of PTC by establishing LAG classification. In fact, the group with a low LAG score consistently indicated an excellent prognosis across various clinical subgroups, demonstrating the potential clinical value of the LAG score.

Lysosomes play a crucial role in regulating cellular degradation, autophagy, antigen presenting and the tumor micro-environment during carcinogenesis process, influencing cancer cell survival, metastasis, and resistance to therapies [10]. Our findings highlighted the significant role of lysosome-related functions in the risk stratification of PTC. Currently, research on the correlation between lysosome and PTC is still insufficient, while there is clue suggesting a close

relationship between the two. PTC is reported to be characterized by the activation of the PI3K/AKT pathway and MAPK pathway, of which the mechanistic Target of Rapamycin Complex 1 (mTORC1) is a described key component [27,28]. The growth-promoting effect of lysosomes is due to their identification as activators of the mTORC1 kinase, thus lysosomal dysfunction may be associated with the initiation and progression of PTC [29]. When both nutrients and growth factors are abundant, the kinase activity of mTORC1 is turned on at the lysosomal membrane [30,31], which in turn generates precursors for amino acid, lipid, and nucleotide synthesis, promoting cell growth [32–34]. mTORC1 has been reported to promote tumorigenesis in PTC by activating tumor-promoting signaling pathways [35,36]. Adding mTORC1 inhibitors to treatments has shown significant antitumor activity [37,38]. However, given the current limited research, the mechanism of lysosomes in PTC still requires further investigation.

We identified *DNASE2B* as a PTC prognosis-related LAG and demonstrated through *in vitro* experiments that *DNASE2B* plays a critical role in PTC cell growth and invasion. As an endonuclease, *DNASE2B* has been reported to be involved in the nuclear envelope breakdown and DNA degradation processes in fibroblasts [39–41]. Mice lacking the *DNASE2B* gene fail to degrade DNA during lens cell differentiation, leading to the accumulation of undigested DNA in fibroblasts and ultimately resulting in cataracts [42]. However, there have been no reports linking *DNASE2B* to tumor-related mechanisms, and only preliminary result suggests an association between *DNASE2B* and chemotherapy resistance [43]. There have been reports of endonucleases affecting redox functions in the thyroid, which may provide a clue as to how *DNASE2B* influences thyroid function and thus tumor prognosis [44].

Our results showed that PTC patients with a high LAG score, indicating a relatively higher risk, had significantly elevated monocyte levels. It has been reported that the lymphocyte-to-monocyte ratio (LMR) has important prognostic value in predicting PTC outcomes [45]. A lower LMR is associated with more aggressive clinicopathological features, such as larger tumor size, lymph node metastasis (LNM), multifocal PTC, and advanced N/M staging, while a higher LMR is significantly associated with better OS, PTC-free survival, and lower recurrence risk in PTC patients [46–48]. Peritumoral infiltration in PTC is also related to LMR, though the specific mechanism remains unclear [49,50]. Our results further suggest the clinical value of monocytes and LMR in predicting PTC prognosis.

There are differences in drug resistance among different LAG groups, which may be related to the pathways identified in the GO enrichment analysis including copper metabolism. Copper is associated with resistance to various chemotherapy drugs [51,52]. It has been reported that copper enhances resistance to genotoxic drugs through DNA damage repair activated by ATOX1 [53]. Copper transport proteins may also contribute to differences in drug resistance [54,55]. Additionally, other enriched biological processes, the regulation of fatty acid metabolism pathways and fatty acid transport proteins may also influence the response to tumor-targeted therapies, which might be a reason how prognostic differences appear [56,57]. Lysosomes have been reported to be associated with chemoresistance through multiple mechanisms, including drug binding and sequestration, regulation of the mitotic cycle, and activation of resistance-related signaling pathways [12,58–60]. However, the GEO database does not contain drug resistance-related data for PTC, the differences in IC50 still provide evidence for the role of lysosomes in chemoresistance in PTC.

Bioinformatics-based scoring holds significant potential in clinical practice by individual risk stratification, treatment response analysis, or prognostic score evaluation [61–63]. The established LAG score enables personalized health management and has the potential of clinical decision-making guidance, such as predicting tumor environment status, optimizing treatment selection, and stratifying patients for clinical trials. In addition, a composite risk stratification and prognostic evaluation system is more accurate than a single system [61–63]. Our results highlight that the present study, together with other PTC biomarkers, may further optimize risk stratification of PTC.

Although studies on lysosomes and related risk stratification are relatively limited in PTC patients, lysosomes have been reported to influence the risk of tumor initiation and progression in many tumor types [64]. LAGs have been implicated in risk stratification across various cancers, including acute myeloid leukemia, hepatocellular carcinoma, and ovarian cancer, suggesting a broad role of lysosomes in tumor biology [65–67]. The influence of LAGs on the immune

microenvironment, which has been widely mentioned among different cancer types, is also worthy of further study. Given the well-established involvement of lysosomes in general cancer-driving signaling pathways such as mTORC1 [29], it would be an interesting topic to explore the function of LAGs consistently across different tumor types.

In conclusion, we established a new LAG scoring system and provided a new potential target for intervention. Limited by database type, bioinformatics results inevitably exist with regional and ethnic bias. The PTC datasets in the GEO database do not provide information on sample survival time or survival status. We only performed stability validation of the model using an internally randomized cohort-splitting algorithm, without validation from an independent dataset. Building upon the new insights into the role of lysosomes in PTC provided in this study, further multicenter and large-sample clinical cohort analyses will be essential in the future to validate the clinical applicability of this model.

## Supporting information

**S1 Table. Baseline characterization of normal and PTC samples in TCGA database.**
(XLSX)

**S2 Table. The gene list of lysosome related genes.**
(XLSX)

**S3 Table. Identification of differentially expressed genes between cell populations.**
(XLSX)

**S1 raw images. Raw images.**
(PDF)

## Author contributions

**Methodology:** Kai Yue.

**Software:** Kai Yue, Chao Jing.

**Validation:** Chao Jing, Xudong Wang.

**Writing – original draft:** Jianhua Zhang.

**Writing – review & editing:** Yansheng Wu, Xudong Wang.

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
