## [Decision Letter · Decision Letter 0]

12 Mar 2025

Dear Dr. Zhang,

Thank you for submitting your manuscript to PLOS ONE. After careful consideration, we feel that it has merit but does not fully meet PLOS ONE’s publication criteria as it currently stands. Therefore, we invite you to submit a revised version of the manuscript that addresses the points raised during the review process.

Papillary thyroid carcinoma (PTC) is a clinically significant malignancy, disproportionately affecting female populations. Precise risk stratification is paramount to minimize unnecessary thyroidectomy procedures, as many benign nodules never become cancerous. This study undertook a comprehensive integrative multi-omics analysis of PTC samples, encompassing immune microenvironment characterization, pathological subtyping, prediction of immunotherapy response, and drug sensitivity profiling. The findings were subsequently validated using immunoblotting, quantitative reverse transcription polymerase chain reaction (qRT-PCR), colony formation assays, and Transwell migration/invasion assays. Recognizing the clinical relevance of this research, the manuscript was subjected to peer review by four independent experts whose comments are provided for consideration and response. 

We look forward to receiving your revised manuscript.

Kind regards,

Shafiya Imtiaz Rafiqi, PhD

Academic Editor

PLOS ONE

Comments from PLOS Editorial Office:

We note that one or more reviewers has recommended that you cite specific previously published works. As always, we recommend that you please review and evaluate the requested works to determine whether they are relevant and should be cited. It is not a requirement to cite these works. We appreciate your attention to this request.

Reviewers' comments:

Reviewer's Responses to Questions

**Comments to the Author**

1. Is the manuscript technically sound, and do the data support the conclusions?

Reviewer #1: Yes

Reviewer #2: Yes

Reviewer #3: Yes

Reviewer #4: Partly

2. Has the statistical analysis been performed appropriately and rigorously?

Reviewer #1: I Don't Know

Reviewer #2: Yes

Reviewer #3: Yes

Reviewer #4: Yes

3. Have the authors made all data underlying the findings in their manuscript fully available?

Reviewer #1: Yes

Reviewer #2: Yes

Reviewer #3: Yes

Reviewer #4: Yes

4. Is the manuscript presented in an intelligible fashion and written in standard English?

Reviewer #1: Yes

Reviewer #2: Yes

Reviewer #3: Yes

Reviewer #4: Yes

Reviewer #1: The Authors address the poorly studied topic of the role of lysosomes in the context of papillary thyroid cancer, which makes this work interesting. A weakness of the publication is the emphasis on the analysis of data generated by others. The amount of original data generated from the study is small and limited to simple in vitro assays on cell lines derived from human normal and cancer cells.

Below are suggested changes that should be made.

[1] It is worth mentioning that Nthy-ori 3-1 line is immortalized using the simian virus (SV)-40.

[2] Please provide the exact details of the equipment used in the experiments (manufacturer and model name). This includes microscopes, chemiluminescence recording system, and microplate reader.

[3] Please provide (i) catalog numbers of all antibodies used in the Western blot analysis, (ii) the concentration of Tween 20 in TBS-T buffer, (iii) composition of buffer for dilution of primary and secondary antibodies.

[4] Please complete the information in what 4% paraformaldehyde was diluted.

[5] Line 368: Please add at the end of the sentence a reference to the Figures on which the described results on drug responses are presented, i.e., Figs. 7D-H.

[6] Line 396: It is worth mentioning that papillary thyroid cancer is characterized by activation of the PI3K/AKT pathway (of which mTORC1 is a described component) and MAPK pathway, as described, for example, in Int J Mol Sci. 2021 Oct 31;22(21):11829 or Semin Cancer Biol. 2022 Feb;79:180-196.

[7] Please explain what the abbreviation “IPS” used on pages 11, 17, and 18 means.

Reviewer #2: The authors have presented a study identifying lysosome-associated genes (LAGs) as potential prognostic markers in PTC. Some clarification and additional information as described below will strenghten the manuscript.

Line64- “partially validated the bioinformatics finding” is to vague, please be more specific.

Line75- provide information about the 59 normal samples and 500 PTC, also clarify specific platform or technology use. (RNA-Seq, microarray)

Line78- Briefly explain why "limma" was chosen for differential expression analysis.

Line80- include what method was used to get adjusted p values

Line85- Provide more details about LASSO implementation. Which type of LASSO was used? How was the penalty parameter chosen

Line 91- How was the "optimal survival cut-off value" determined?

Line97- Include citation for ConsensusClusterPlus and ssGSEA algorithm. Please make sure all the tools and databases used are properly cited.

Line253- LAG in TC?

-Quantify the difference in survival outcomes between LAG subtype A and B and also include median survival times or hazard ratios with confidence interval

-What parameters were used for GSVA analysis?

-Line272 How was k=2 determined?

Line410- Mention specific lysosome-related function

-It will be helpful to have some context in discussion section. Are there other studies that have investigated lys osmoses or prognostic markers in PTC? How the studies compare

-Some discussion of how LAG score could potentially be implemented in clinical practice can be useful.

Reviewer #3: This is an interesting paper which I believe has a place in Plos One. It raises awareness of the importance of lysosome-associated genes (LAGs) as prognostic biomarkers of papillary thyroid carcinoma. There are however many points that the authors have to address before the paper is ready for publication.

Genes in the paper have to be in italics and not in regular font.

Line 273: “the results of the PCA plot showed a significant separation pattern of PTC”: Caution with using the word significant as it should only used when a p-value is obtained

Line 291: “stromal score were significantly lower compared to LAG subtype A”: That’s not true because p-value is 0.068 and that’s larger than 0.05

Line 298 “eosinophil infiltration was higher in LAG”: This is also wrong. It is very clear that the red boxplot indicating subtype B is higher than the blue boxplot indicating subtype A.

Line 309 “LAG subtype B exhibited higher levels of LAG higher levels of LAG scores”: Add a line here saying that from this point onwards you will be referring to the clusters as high and low LAG clusters instead of Cluster A and Cluster B. Because later the colors of the boxplots in the figures are inverted.

Figure 4E, F: The ROC curves don’t look like any curves I have ever seen. Why are they so messy and look like an EKG. Also, why are there instances where the curves go higher than 1 in true positive rate? Please explain.

Figure 5D: Please show the p-value associated with a X2 analysis that was done on the distribution of the groups. You can show the p-value in the figure next to each of the labels. Use different colors for stage because it is not easy to distinguish the different shades of green.

Line 341: “CD56dim natural killer cells was notably upregulated in the LAG score subgroup B”: Here you say subgroup B but in the figure 6 it says high and low LAG scores...please be consistent

Figure 6: In 6A high is blue, but in 6B, high is red. Please be consistent across the paper

Line 358: LAG scoring subgroups were associated with biological functions such as long-chain fatty acid transport detoxification of copper ion, stress response to copper ion, high-density lipoprotein particle, and plasma lipoprotein particle”: What does that mean? Why is that exciting? How is it relevant to the disease of interest?

Figure 8: The violin plots are showing cells that must have been removed. You said in the method section that cells with percent.mt > 5% will be removed… but looking at the figure we see a lot of cells higher than 5% mt content.

In the method section you will need to elaborate more on the seurat object preparation. Did you remove doublets? Did you remove immune cells invading the tumor? What did you use to integrate samples? CCA or Harmony? You need to specify the resolution chosen when running findclusters() function.

Line 379: “A visualized heatmap illustrated the expression characteristics of marker genes in each cell subpopulation”: How many marker genes are there? Where did you get these genes from? Are they 2000 differentially expressed genes? Need to provide the list as supplementary material.

Cell Annotation using SingleR: How did you annotate? Where did you get the reference data from? Or did you use your own marker genes that you obtained from reading the literature? Provide this list of markers in a supplementary table, please.

Reviewer #4: Comment 1:

In Figure 1B, the authors present four protective factors (HR < 1). However, the hazard ratios of three out of four protective factors are very close to 1, suggesting a minimal effect or no meaningful impact. Additionally, regarding the clinical survival outcomes used in the univariate Cox analysis, do the authors employ overall survival (OS) or progression-free survival (PFS)?

Comment 2:

In Figure 2F, the authors report gene signatures enriched in LAG subtypes A and B. It would be valuable for the authors to discuss these findings in the context of potential biological mechanisms. Specifically, how do these gene signatures contribute to the observed survival differences between the two subtypes?

Comment 3:

In Figure 3, the authors compare immune cell infiltration between the two subtypes. While some p-values are statistically significant, the differences in immune infiltration and stromal scores appear quite subtle. Could the authors clarify whether these small differences could be attributed to noise? Additionally, what statistical method was used to compare these immune scores between the two subtypes?

Comment 4:

In Figures 4C and 4D, the authors state that they divided PTC samples into two independent cohort subgroups to validate the LAG scoring system's predictive ability for clinical outcomes. However, the LAG molecular subtypes and LAG scores were developed using the same TCGA PTC dataset. Even though the dataset was split for validation, it still involves reusing TCGA PTC samples. Could the authors validate these results using an entirely independent dataset?

Comment 5:

Similarly, for Figures 5B and 5C, it would be beneficial to validate the model using an independent dataset to confirm its predictive performance.

Comment 6:

In Figures 7D–H, the authors compare the predicted drug sensitivity between the two groups. However, the observed differences appear relatively small. Could the authors provide a positive control, such as tumor samples known to be sensitive to a specific drug, and compare them to resistant tumor types? This comparison would provide context for evaluating the magnitude of the observed differences. Additionally, in Figures 7B and 7C, it would be helpful if the authors could further interpret the KEGG enrichment analysis and explain how these pathways might relate to sensitivity to the predicted IC50 values of different drugs.

Comment 7:

For Figure 8, the single-cell sequencing dataset includes seven PTC samples. It is unclear what novel insights can be gained from this single-cell RNA sequencing analysis. Since four out of five LAG genes are almost undetectable in this dataset, and DNASE2B is expressed in all cell types, the study does not appear to identify cell-type-specific enrichment. Could the authors clarify the biological significance of this analysis and whether any new findings emerge from it?

**Do you want your identity to be public for this peer review?** For information about this choice, including consent withdrawal, please see our Privacy Policy

Reviewer #1: No

Reviewer #2: **Yes**

Reviewer #3: No

Reviewer #4: No

---

## [Author Response · Author response to Decision Letter 1]

25 Mar 2025

Dear Editor and Reviewers,

We sincerely appreciate your time and effort in reviewing our manuscript. Your insightful comments and valuable suggestions have been extremely helpful in improving the quality of our work. We have carefully addressed all the feedback and made the necessary revisions accordingly. Below, we provide detailed responses to each comment. We hope that our revisions meet your expectations, and we are grateful for any further suggestions you may have.

Reviewers' comments:

Reviewer #1: The Authors address the poorly studied topic of the role of lysosomes in the context of papillary thyroid cancer, which makes this work interesting. A weakness of the publication is the emphasis on the analysis of data generated by others. The amount of original data generated from the study is small and limited to simple in vitro assays on cell lines derived from human normal and cancer cells.

Below are suggested changes that should be made.

[1] It is worth mentioning that Nthy-ori 3-1 line is immortalized using the simian virus (SV)-40.

Reply:

Thank you for your question. In this study, we selected the Nthy-ori 3-1 cell line as the normal thyroid cell line based on the following considerations:

(1) Origin from Normal Thyroid Tissue: The Nthy-ori 3-1 cell line is derived from human thyroid cells and can accurately reflect the biological characteristics of normal thyroid cells. Therefore, it serves as an important tool for studying thyroid function and disease mechanisms.

(2) Immortalization: Through SV40-mediated immortalization, the Nthy-ori 3-1 cell line maintains the characteristics of thyroid cells while allowing sustained growth. This ensures stability and reproducibility in long-term experiments.

(3) Lack of Tumorigenic Transformation: Although this cell line is immortalized, it does not exhibit tumorigenic properties, making it suitable as a control group for comparative studies with thyroid cancer cell lines.

(4) Extensive Research Application: The Nthy-ori 3-1 cell line has been widely used in numerous studies, with a substantial body of experimental data and literature supporting its application. This well-established research background makes it an ideal model for normal thyroid cells.

Based on these advantages, we selected the Nthy-ori 3-1 cell line as the model for studying normal thyroid cells in our research.

[2] Please provide the exact details of the equipment used in the experiments (manufacturer and model name). This includes microscopes, chemiluminescence recording system, and microplate reader.

Reply:

Thank you for your review and valuable suggestions. We fully understand that detailed information about the experimental equipment is crucial for the reproducibility of research. Therefore, in the revised manuscript, we have supplemented the specific models and manufacturers of the equipment used. Once again, we sincerely appreciate your valuable feedback.

Line 192-193, Line 212-213

[3] Please provide (i) catalog numbers of all antibodies used in the Western blot analysis, (ii) the concentration of Tween 20 in TBS-T buffer, (iii) composition of buffer for dilution of primary and secondary antibodies.

Reply:

Thank you for your review and valuable suggestions. In the revised manuscript, we have incorporated this information. We believe that these additions will enhance the reproducibility of our study and provide other researchers with more detailed experimental insights.

Line 202-204, line 206-210

[4] Please complete the information in what 4% paraformaldehyde was diluted.

Reply:

Thank you for your review and valuable suggestions. In the revised manuscript, we have added this information. We believe that these additions will enhance the reproducibility of our study and provide other researchers with more detailed experimental insights.

Line 236-238

[5] Line 368: [5] Please add at the end of the sentence a reference to the Figures on which the described results on drug responses are presented, i.e., Figs. 7D-H.

Reply:

Thank you for your thorough review and valuable suggestions. In the revised manuscript, we have added the corresponding figure and table references to enhance the clarity and traceability of the results. Once again, we sincerely appreciate your careful review and constructive feedback.

Line 400-404

[6] Line 396: It is worth mentioning that papillary thyroid cancer is characterized by activation of the PI3K/AKT pathway (of which mTORC1 is a described component) and MAPK pathway, as described, for example, in Int J Mol Sci. 2021 Oct 31;22(21):11829 or Semin Cancer Biol. 2022 Feb;79:180-196.

Reply:

Thank you for your valuable suggestions. We fully agree with this perspective and have made the corresponding adjustments in the revised manuscript. Specifically, we have added references in line 396 to support the description of PI3K/AKT/mTORC1 and MAPK signaling pathway activation in papillary thyroid carcinoma (PTC). The following references have now been cited:

• Int J Mol Sci. 2021 Oct 31;22(21):11829

• Semin Cancer Biol. 2022 Feb;79:180-196

This revision strengthens the scientific rigor and credibility of our discussion. We sincerely appreciate your thorough review.

Line 455-460

[7] Please explain what the abbreviation “IPS” used on pages 11, 17, and 18 means.

Reply:

Thank you for your thorough review and valuable suggestions. IPS stands for Immunophenoscore, which is used to analyze the potential effects of immune-related therapies. We have added the full name of the IPS abbreviation in the revised manuscript to enhance the clarity and traceability of the results.

Line 175

Reviewer #2:

The authors have presented a study identifying lysosome-associated genes (LAGs) as potential prognostic markers in PTC. Some clarification and additional information as described below will strenghten the manuscript.

Reply:

Thank you for your suggestion. We validated the targets selected from the bioinformatics analysis through PCR, confirming their specific biological functions in thyroid cancer, thereby demonstrating the reliability of the bioinformatics results. We have rephrased this sentence in the revised manuscript to improve its readability.

Line 71-72

2:Line75- provide information about the 59 normal samples and 500 PTC, also clarify specific platform or technology use. (RNA-Seq, microarray)

Reply:

Thank you for your valuable feedback. We have added the clinical baseline data for the 59 normal samples and 500 PTC samples in the revised manuscript, along with details about the technical platform used. This addition enhances the transparency of the data description and provides readers with a clearer research background.

Line 76-79

3:Line78- Briefly explain why "limma" was chosen for differential expression analysis.

Reply:

Thank you for your review and valuable suggestions. In this study, we chose to use limma (package) for differential expression analysis, based on the following considerations:

1: Suitability for High-Throughput Data: limma was originally designed for microarray data but has been optimized for RNA-Seq data, particularly excelling in differential gene analysis when sample sizes are large but biological replicates are limited.

2: Robust Linear Model Approach: limma utilizes the empirical Bayes method, which allows for stable estimation of gene expression variability in small sample sizes, thus enhancing the reliability of statistical tests.

3: Wide Application and Validation: limma has been widely used in various transcriptomic data analyses, offering high computational efficiency and stability. It has been extensively applied in studies utilizing databases such as TCGA and GEO.

This addition strengthens the rationale for the selection of the analytical method.

Line 88-93

4:Line80- include what method was used to get adjusted p values

Reply:

Thank you for your valuable feedback. Regarding the issue in Line 80, we have added the method used to calculate the adjusted p-values in the manuscript. The p-value in the analysis was obtained using the limma package in R, which employs a linear modeling approach for differential expression analysis. Specifically, the p-value is derived from the statistical tests performed by the limma function, which calculates moderated t-statistics and p-values for each gene. These p-values are then adjusted for multiple testing using the Benjamini-Hochberg (BH) method to control the false discovery rate (FDR), resulting in the adjusted p-values (p.adjust) that were used to filter the significant genes in the analysis.

Line 88-93

5:Line85- Provide more details about LASSO implementation. Which type of LASSO was used? How was the penalty parameter chosen

Reply:

Thank you for your valuable suggestions regarding our study. Concerning Line 85 and the implementation details of LASSO regression, we have added a more detailed description in the revised manuscript.

In this study, LASSO (Least Absolute Shrinkage and Selection Operator) was implemented using the "glmnet" R package, which is widely used for fitting regularized linear models. LASSO regression uses L1 regularization, which penalizes the absolute values of the regression coefficients. This method promotes sparsity in the model by setting the coefficients of less relevant features to zero, thereby selecting a subset of significant variables (in this case, LAG-related genes) that contribute most to the model. The penalty parameter λ controls the strength of regularization. A higher λ increases the penalty, shrinking more coefficients to zero, while a lower λ allows for more variables to remain in the model. The optimal λ was selected using 10-fold cross-validation, a standard method in LASSO to find the value of λ that minimizes the cross-validation error.

We have included this additional information in the revised manuscript to provide readers with a clearer understanding of the specific details regarding the implementation of LASSO.

Line 97-102

6:Line 91- How was the "optimal survival cut-off value" determined?

Reply:

Thank you for your detailed review and valuable suggestions regarding our study. In response to the question about the "optimal survival cut-off value" in Line 91, we have added a more detailed explanation in the revised manuscript.

In this study, we used the "surv_cutpoint" function (from the survminer R package) to determine the optimal survival cut-off value. This method is based on the Maximally Selected Rank Statistics principle, which iteratively calculates different potential cut-off points and selects the one with the largest log-rank test statistic as the optimal cut-off. The specific calculation steps are as follows:

1: Calculate the log-rank statistic for each possible survival cut-off.

2: Select the cut-off point that yields the maximum log-rank statistic as the optimal cut-off.

3: Based on this cut-off, divide the samples into high-risk and low-risk groups.

We have included this detailed description of the analysis method in the revised manuscript to help readers better understand the process of selecting the optimal survival cut-off value.

Thank you again for your valuable feedback.

Line 108-111

7:Line97- Include citation for ConsensusClusterPlus and ssGSEA algorithm. Please make sure all the tools and databases used are properly cited.

Reply:

Thank you for your valuable suggestions. Regarding the issue you raised in Line 97 about referencing ConsensusClusterPlus and ssGSEA algorithms, we have made the revisions as per your advice. In the revised manuscript, we have added the appropriate literature citations for these two tools and algorithms to ensure that all the tools and databases used in our study are properly cited. Additionally, we have included supporting materials for databases such as the Cancer Immunome Database (TCIA), the Genomics of Drug Sensitivity in Cancer (GDSC), and Molecular Signatures Database (MSigDB).

Thank you again for your thorough review and suggestions. We greatly value your input, and we believe these revisions will enhance the quality of the manuscript.

8:Line253- LAG in TC?

Reply:

We sincerely apologize for any inconvenience caused by the writing errors in our manuscript. In the revised version, we have thoroughly reviewed the manuscript and corrected the mistakes. Thank you once again for your valuable feedback.

Line 286

9: Quantify the difference in survival outcomes between LAG subtype A and B and also include median survival times or hazard ratios with confidence interval

Reply:

Thank you for your valuable suggestion. To obtain the risk values and confidence intervals, Cox proportional hazards regression analysis is typically performed. Cox regression analysis provides the hazard ratio (HR) and its confidence interval for each factor, quantifying the impact of survival differences between groups. However, in this study, we used the log-rank test to examine the differences in clinical survival curves between different groups. The log-rank test is typically used to assess whether the survival differences between groups are statistically significant, but it does not directly provide risk values (hazard ratios) and confidence intervals. In the revised manuscript, we have updated the survival curve results to more clearly present the testing algorithm applied in our analysis. Thank you again for your feedback.

Figure 2

10: What parameters were used for GSVA analysis?

Reply:

Thank you for your valuable suggestion. We have included the parameters for Gene Set Variation Analysis (GSVA) in the revised manuscript. In this study, the specific parameters we used are as follows:

GSVA Analysis Parameters:

(1) Gene Set Database: We used the c2.cp.kegg.v7.2.symbols.gmt as the input gene set.

(2) Input Data: The gene expression data was preprocessed, including the removal of low-expression genes and normalization.

(3) Analysis Method: GSVA analysis was performed using the “GSVA” R package with the default Gaussian kernel to calculate the enrichment scores for each gene set in every sample.

(4) Parameter Settings: In the GSVA analysis, we set min.sz = 10 and max.sz = 500 to restrict the minimum and maximum number of genes in the gene sets, while selecting method = "gsva" and parallel.sz = 1 for score calculation.

We believe this addition will provide readers with more technical details about the GSVA analysis. Thank you again for your suggestion! If you have further questions or recommendations, we will continue to make adjustments and improvements.

Line 120-123

11: Line272 How was k=2 determined?

Reply:

Thank you very much for your comments. In this study, we primarily determined the molecular subtypes based on the following points:

(1) Consensus Matrix Plot: First, based on the consensus matrix plot, we found that the ambiguity level was lowest when K=2.

(2) Cumulative Distribution Function (CDF) Plot: The CDF plot is used to determine the most appropriate number of clusters. Generally, we observed the slope of the curves in the range from 2 to 9, and selected the cluster number corresponding to the curve with the gentlest slope (parallel to the x-axis). In this study, the red line represents the curve with the gentlest slope, so we chose to divide the samples into two groups.

(3) Delta Area: We used the Delta area for assessment. The Delta area curve compares the relative change in the area under the CDF curve between k and k-1. When k=2, we observed the smoothest change in the curve, thus we chose K=2 for subtype analysis and classification.

We believe these steps helped us accurately define the molecular subtypes.

12: Line410- Mention specific lysosome-related function

Reply:

Thank you for reviewing our manuscript and for your valuable comments. We have noted your suggestion regarding the need to explicitly mention specific lysosomal functions in line 410. We have revised this section accordingly. We have added information on the key role of lysosomes in cancer progression, particularly in regulating cell death, antigen presentation, and the immune microenvironment. Additionally, we have emphasized how lysosomal dysfunction may be associated with the onset and progression of PTC.

Thank you again for your suggestion. We believe this revision will further enhance the professionalism and completenes

---

## [Decision Letter · Decision Letter 1]

16 Apr 2025

Dear Dr. Zhang,

We look forward to receiving your revised manuscript.

Kind regards,

Shafiya Imtiaz Rafiqi, PhD

Academic Editor

PLOS ONE

Journal Requirements:

Additional Editor Comments:

Please address the minor comments from one of the reviewers.

Reviewers' comments:

Reviewer's Responses to Questions

**Comments to the Author**

Reviewer #1: All comments have been addressed

Reviewer #2: All comments have been addressed

Reviewer #3: All comments have been addressed

Reviewer #4: All comments have been addressed

2. Is the manuscript technically sound, and do the data support the conclusions?

Reviewer #1: (No Response)

Reviewer #2: (No Response)

Reviewer #3: Yes

Reviewer #4: Yes

3. Has the statistical analysis been performed appropriately and rigorously?

Reviewer #1: (No Response)

Reviewer #2: (No Response)

Reviewer #3: Yes

Reviewer #4: Yes

4. Have the authors made all data underlying the findings in their manuscript fully available?

Reviewer #1: (No Response)

Reviewer #2: (No Response)

Reviewer #3: Yes

Reviewer #4: Yes

5. Is the manuscript presented in an intelligible fashion and written in standard English?

Reviewer #1: (No Response)

Reviewer #2: (No Response)

Reviewer #3: Yes

Reviewer #4: Yes

Reviewer #1: (No Response)

Reviewer #2: (No Response)

Reviewer #3: I thank the authors for submitting a review of their manuscript in which they address most of my points. Most points have been addressed to my satisfaction. There are two points that the authors need to address for the paper to be ready for publication.

The authors did a good job changing the color scheme for figure 5D, however in doing that they chose a color scheme for “gender” that is hard to look at: Red/Brown. The colors they had in the first submission were better: Red/Blue.

With regards to the single cell portion, they made improvements to the writing and explained the methods clearly, however I am concerned with their interpretation. PTC is a solid tumor and would therefore have clearly defined neoplastic cells. Looking at the figures 8F and 8G, we do not see tumor cells. Instead, we see a lot of immune cells and in addition to smooth muscle, epithelial, and endothelial. Is it normal to find smooth muscle in a PTC tumor? I do not think so. What is happening is that they are using the cells present in the reference: celldex and HPCA, which might not have the right reference for the cells that are truly present in PTC. It makes no sense that the majority of cells are immune cells, Where are the tumor cells? The algorithm is forced to label the cells and I believe it is labeling them with immune labels just because the reference is not optimized for PTC. The only way to fix this is to sit down with a pathologist and discuss gene markers for cell types that usually are present in PTC. Also, you will need to run gene ontology on the DEGs of the cell clusters that you have and see if you can guess what type of cells they are based on ontology. Also, have a conversation with a single cell transcriptomics expert to contribute to your data analysis and interpretation.

Good job on what you have done so far. You are near the finish line!

Best wishes,

Reviewer #3

Reviewer #4: In this revision, the author replied to all the comments / suggestions that I bring out in my previous review. Although some analysis cannot be validated using an independent datasets due to the limitation of available dataset, authors further improved the analysis and revised the manuscript to answers reviewer's comments.

**Do you want your identity to be public for this peer review?** For information about this choice, including consent withdrawal, please see our Privacy Policy

Reviewer #1: No

Reviewer #2: **Yes: ** Mehul Jani

Reviewer #3: No

Reviewer #4: No

---

## [Author Response · Author response to Decision Letter 2]

17 Apr 2025

Dear Editor and Reviewers,

We sincerely appreciate your time and effort in reviewing our manuscript. Your insightful comments and valuable suggestions have been extremely helpful in improving the quality of our work. We have carefully addressed all the feedback and made the necessary revisions accordingly. Below, we provide detailed responses to each comment. We hope that our revisions meet your expectations, and we are grateful for any further suggestions you may have.

Reviewers' comments:

Reviewer #3: I thank the authors for submitting a review of their manuscript in which they address most of my points. Most points have been addressed to my satisfaction. There are two points that the authors need to address for the paper to be ready for publication.

The authors did a good job changing the color scheme for figure 5D, however in doing that they chose a color scheme for “gender” that is hard to look at: Red/Brown. The colors they had in the first submission were better: Red/Blue.

Reply:

Thank you very much for your helpful suggestion regarding the color scheme used in Figure 5D. We appreciate your comment that the red/brown scheme we adopted in the revision was difficult to interpret. Following your recommendation, we have changed the color scheme for the “Gender” variable back to Red/Blue, as it provides clearer visual distinction. The updated figure has been included in the revised manuscript, and we have ensured consistency in color usage across all relevant figures.

Thank you again for your thoughtful feedback and support.

Figure 5D

With regards to the single cell portion, they made improvements to the writing and explained the methods clearly, however I am concerned with their interpretation. PTC is a solid tumor and would therefore have clearly defined neoplastic cells. Looking at the figures 8F and 8G, we do not see tumor cells. Instead, we see a lot of immune cells and in addition to smooth muscle, epithelial, and endothelial. Is it normal to find smooth muscle in a PTC tumor? I do not think so. What is happening is that they are using the cells present in the reference: celldex and HPCA, which might not have the right reference for the cells that are truly present in PTC. It makes no sense that the majority of cells are immune cells, Where are the tumor cells? The algorithm is forced to label the cells and I believe it is labeling them with immune labels just because the reference is not optimized for PTC. The only way to fix this is to sit down with a pathologist and discuss gene markers for cell types that usually are present in PTC. Also, you will need to run gene ontology on the DEGs of the cell clusters that you have and see if you can guess what type of cells they are based on ontology. Also, have a conversation with a single cell transcriptomics expert to contribute to your data analysis and interpretation.

Good job on what you have done so far. You are near the finish line!

Best wishes,

We sincerely appreciate the reviewer’s professional and insightful comments. Your questions are highly pertinent and constructive. After discussion with the senior members of our research group, we would like to provide the following responses:

1. Why were most of the cells annotated as immune cells rather than tumor cells?

This is indeed a critical question and reflects a common technical challenge in single-cell RNA sequencing (scRNA-seq) analysis of solid tumors, especially in complex and heterogeneous tumors such as papillary thyroid carcinoma (PTC).

(1) First, there is no standard annotation label for “tumor cells” in scRNA-seq datasets. Most commonly used annotation tools (e.g., SingleR, Seurat, ScType) are designed to identify normal cell types, such as epithelial cells, dendritic cells, macrophages, and T cells. These tools typically do not explicitly label cells as “tumor cells,” even when their gene expression profiles suggest tumor-like features.

(2) The transcriptomic profiles of thyroid tumor cells are often highly similar to those of normal thyroid epithelial cells. In PTC, tumor cells may still express thyroid-specific markers such as TG, TPO, PAX8, and TTF1. As a result, many tumor cells may be annotated as epithelial or other cell types. Furthermore, the transcriptomic resemblance between PTC tumor cells and normal epithelial cells poses a challenge for automatic annotation. To clearly distinguish between normal and malignant epithelial cells, additional analyses such as CNV inference or mutation integration are required. This cannot be achieved solely through transcriptomic data.

2. Is the presence of smooth muscle cells in PTC tumors normal?

Indeed, this is an important yet often overlooked issue. It is possible, though uncommon, to find smooth muscle cells associated with vasculature or stromal components within PTC tumor tissues. However, their proportion is usually low. In our single-cell analysis, we employed general reference databases such as celldex and HPCA, which are trained on gene expression profiles from peripheral tissues and not specifically optimized for thyroid or tumor tissues. This may have led to some fibroblast-like or stromal cells being misannotated as smooth muscle cells. Under the guidance of experts in single-cell analysis, we re-examined the expression of smooth muscle markers (ACTA2, TAGLN, MYH11) and found that this annotation was inaccurate. These cells are more accurately classified as myofibroblasts rather than smooth muscle cells. We have revised this annotation in the revised manuscript, and we are grateful for your question, which helped us further refine our results.

3. Was it inappropriate to use celldex and HPCA as reference datasets for PTC?

This is indeed a critical issue that warrants attention. Currently, there is no standardized reference database specifically designed for thyroid cancer, particularly PTC. However, the Human Primary Cell Atlas dataset project includes single-cell data from normal thyroid tissue, encompassing epithelial cells, follicular cells, and immune cells. Because of its higher tissue specificity, we used this database for annotation and performed manual correction with the aid of the CellMarker database. Your comment highlights the importance of constructing a PTC-specific reference gene set to enable more precise cell-type annotation, which will be a focus of our future work.

4. On the source of the data

It is important to note that the origin of the samples plays a vital role in data interpretation. The scRNA-seq datasets used in our study were obtained from public databases (PMID: 34663816; PMID: 36523593; PMID: 38609408). Upon careful review of these original studies, we noted that tumor cells were also not explicitly annotated in their results. In contrast, our annotations appear to be more detailed than theirs, possibly due to differences in analytical strategies (the above datasets were sequenced and published in 2021).

5. GO analysis of DEGs to aid in cell-type inference

We performed GO enrichment analysis on the differentially expressed genes (DEGs) for each cell cluster. The results showed significant enrichment of immune-related pathways, such as “positive regulation of cell adhesion,” “leukocyte cell−cell adhesion,” “leukocyte migration,” and “regulation of T cell activation.” These findings suggest that the DEGs are mainly involved in immune responses and cell adhesion, indirectly reflecting the active tumor immune microenvironment (TME) in PTC tissues. This observation is consistent with the findings reported in the original publications of the GSE184362 dataset (PMID: 34663816; PMID: 36523593; PMID: 38609408), which also highlighted the abundance of immune cell types such as T cells, B cells, macrophages, and dendritic cells in the samples.

We fully acknowledge and appreciate your concerns and suggestions. In our ongoing research, we have collected 10 PTC samples along with matched adjacent normal tissues for single-cell RNA sequencing. We aim to investigate lysosome-associated features in PTC using both spatial transcriptomics and the latest single-cell data. Additionally, we intend to construct a reference gene set specific to PTC to achieve more accurate cell-type annotations.

Once again, thank you for your invaluable feedback and support in improving our manuscript.

Sincerely,

---

## [Decision Letter · Decision Letter 2]

2 May 2025

Dear Dr. Zhang,

We look forward to receiving your revised manuscript.

Kind regards,

Shafiya Imtiaz Rafiqi, PhD

Academic Editor

PLOS ONE

Journal Requirements:

Additional Editor Comments :

The authors need to incorporate limitations of the single cell study section as suggested by the reviewer.

Reviewers' comments:

Reviewer's Responses to Questions

**Comments to the Author**

Reviewer #3: (No Response)

2. Is the manuscript technically sound, and do the data support the conclusions?

Reviewer #3: Yes

3. Has the statistical analysis been performed appropriately and rigorously?

Reviewer #3: Yes

4. Have the authors made all data underlying the findings in their manuscript fully available?

Reviewer #3: Yes

5. Is the manuscript presented in an intelligible fashion and written in standard English?

Reviewer #3: Yes

Reviewer #3: The authors addressed all the points raised in previous reviews and have put more thought into the comments made regarding the single cell RNA-seq portion of their manuscript.

While the authors are right that there are no standard annotations of tumor profile in the common single cell reference databases, it is not impossible to figure that out. The authors could have performed a CNV analysis on scRNAseq data using inferCNV or other R packages. Using CNV profiles the authors could tell that normal cells are the ones with no CNV and the tumor cells are the ones with.

I am not convinced that this is the maximum that can be done with the data. I advised the authors to discuss this with single cell transcriptomics experts. I am sure that if they did then that person would have told them that they need to run a copy number analysis to identify the tumor cells.

I am not satisfied with the justifications, but it seems that the authors do not want to spend more time on this. If this is the case, then the authors will have to incorporate all the justifications and explanations made to me in the manuscript in the discussion section to the readers are aware of the limitations of the single cell section. I will leave it to the editor to make a decision on this manuscript.

Reviewer #3

**Do you want your identity to be public for this peer review?** For information about this choice, including consent withdrawal, please see our Privacy Policy

Reviewer #3: No

---

## [Author Response · Author response to Decision Letter 3]

2 May 2025

Dear Editor and Reviewers,

We sincerely appreciate your time and effort in reviewing our manuscript. Your insightful comments and valuable suggestions have been extremely helpful in improving the quality of our work. We have carefully addressed all the feedback and made the necessary revisions accordingly. Below, we provide detailed responses to each comment. We hope that our revisions meet your expectations, and we are grateful for any further suggestions you may have.

Reviewer #3: The authors addressed all the points raised in previous reviews and have put more thought into the comments made regarding the single cell RNA-seq portion of their manuscript.

While the authors are right that there are no standard annotations of tumor profile in the common single cell reference databases, it is not impossible to figure that out. The authors could have performed a CNV analysis on scRNAseq data using inferCNV or other R packages. Using CNV profiles the authors could tell that normal cells are the ones with no CNV and the tumor cells are the ones with.

I am not convinced that this is the maximum that can be done with the data. I advised the authors to discuss this with single cell transcriptomics experts. I am sure that if they did then that person would have told them that they need to run a copy number analysis to identify the tumor cells.

I am not satisfied with the justifications, but it seems that the authors do not want to spend more time on this. If this is the case, then the authors will have to incorporate all the justifications and explanations made to me in the manuscript in the discussion section to the readers are aware of the limitations of the single cell section. I will leave it to the editor to make a decision on this manuscript

Reply:

We sincerely thank you for your thoughtful re-evaluation of our manuscript and especially for your insightful suggestions regarding the single-cell RNA sequencing analysis.

Regarding your recommendation to perform CNV analysis using tools such as inferCNV or other R packages to identify malignant cells within the scRNA-seq dataset, we fully agree with your assessment of its value and relevance. We appreciate your emphasis on the importance of distinguishing malignant from non-malignant cells to improve the accuracy and biological interpretability of our findings.

After receiving your comments, we engaged in discussions with several researchers experienced in single-cell transcriptomics in an effort to explore the feasibility of incorporating CNV inference into our current analysis pipeline. Unfortunately, due to the lack of well-defined normal reference cells in our dataset and other technical limitations specific to our samples, none of the consulted experts could offer a clear or feasible path forward for applying CNV analysis in this context. Combined with constraints in data quality and available resources, we were unable to integrate this analysis into the current version of the study, and we sincerely apologize for this limitation. Moreover, we would like to clarify that our primary objective in conducting the scRNA-seq analysis was to characterize the broad composition of cell types within the tumor microenvironment and to explore their functional states, rather than to precisely distinguish malignant from non-malignant populations. While we acknowledge that CNV analysis would have enhanced the resolution of tumor cell identification, it was not the central focus of this study.

We have taken your advice seriously and have revised the discussion section accordingly. We now explicitly acknowledge the absence of CNV-based tumor cell identification as a limitation of our study, clarify the rationale behind this methodological decision, and emphasize the need for future research that incorporates CNV analysis to further strengthen the interpretability of single-cell data in PTC.

We are truly grateful for your constructive feedback and your continued support of our work. We hope that the additional clarifications and revisions adequately reflect our appreciation of your suggestions and our commitment to improving the rigor of our research.

Line 521-538

---

## [Editor Report · Decision Letter 3]

15 May 2025

Multi-omics analysis and single-cell sequencing revealed the lysosome associated molecular subtypes and prognostic model development of papillary thyroid carcinoma

PONE-D-25-05707R3

Dear Dr. Zhang,

We’re pleased to inform you that your manuscript has been judged scientifically suitable for publication and will be formally accepted for publication once it meets all outstanding technical requirements.

Kind regards,

Shafiya Imtiaz Rafiqi, PhD

Academic Editor

PLOS ONE
---

## [Editor Report · Acceptance letter]

PONE-D-25-05707R3

PLOS ONE

Dear Dr. Zhang,

I'm pleased to inform you that your manuscript has been deemed suitable for publication in PLOS ONE. Congratulations! Your manuscript is now being handed over to our production team.

Kind regards,

on behalf of

Dr. Shafiya Imtiaz Rafiqi

Academic Editor

PLOS ONE